# Semantic relatedness retroactively boosts memory and promotes memory interdependence across episodes

**James W Antony[1,2]\*, America Romero[2], Anthony H Vierra[2], Rebecca S Luenser[2], Robert D Hawkins[3], Kelly A Bennion[2]**

[1]Center for Neuroscience, University of California, Davis, Davis, United States; [2]Department of Psychology and Child Development, California Polytechnic State University, San Luis Obispo, United States; [3]Princeton Neuroscience Institute, Princeton University, Princeton, United States

**Abstract** Two fundamental issues in memory research concern when later experiences strengthen or weaken initial memories and when the two memories become linked or remain independent. A promising candidate for explaining these issues is semantic relatedness. Here, across five paired-associate learning experiments (N=1000), we systematically varied the semantic relatedness between initial and later cues, initial and later targets, or both. We found that learning retroactively benefited long-term memory performance for semantically related words (vs. unshown control words), and these benefits increased as a function of relatedness. Critically, memory dependence between initial and later pairs also increased with relatedness, suggesting that pre-existing semantic relationships promote interdependence for memories formed across episodes. We also found that modest retroactive benefits, but not interdependencies, emerged when subjects learned via studying rather than practice testing. These findings demonstrate that semantic relatedness during new learning retroactively strengthens old associations while scaffolding new ones into well-fortified memory traces.

\*For correspondence:
james.ward.antony@gmail.com

**Competing interest:** The authors declare that no competing interests exist.

## Editor's evaluation

The study addresses a classical question of the complex dynamics of long term (semantic) memory and episodic learning, using a impressive behavioral data set, revealing the specific interactive patterns between old and new memories. It should have broad implications to how we study learning and memory in general.

## Introduction

When a novice bartender is first learning to make cocktails, they are faced with an intimidating repertoire of closely related recipes. For example, they may begin by learning that a 'Manhattan' is made with *sweet vermouth*. Later, they may learn that a 'martini' is made with *dry vermouth*. These memories are not necessarily independent: the bartender may find that learning to make the martini has retroactively affected their memory of the Manhattan, either by weakening it, a phenomenon known as retroactive interference (*RI*), or by strengthening it, a phenomenon known as retroactive facilitation (*RF*). What properties of the earlier and later memories determine this relationship?

Here, we evaluate an over 70-year-old proposal by Osgood (*Osgood, 1949*) that this relationship depends on *semantic relatedness*. In building up to this proposal, we will consider three broad possibilities. The first possibility is that relatedness has no effect on episodic memory. A second possibility

is that relatedness across experiences introduces interference between memories. Third, relatedness could trigger reminders of prior information during new learning, causing the rehearsal and strengthening of prior memories.

In order for the first (null) account to be correct, memories must be formed distinctly, and processes operating during both encoding and retrieval must be able to accurately pinpoint and isolate memories without any residual effects or dependence on semantics. We consider this account helpful to consider because these processes clearly *are* affected by semantic relatedness, as many decades of research have shown. For instance, semantic relatedness improves memory when both items of a paired-associate are related (*Lyon, 1914*; *Nelson et al., 1997*; *Bein, 2015*), it provides an organizational scaffold for clustering responses during free recall (*Bousfield, 2010*; *Lohnas et al., 2015*; *Talmi and Moscovitch, 2004*; *Irish and Piguet, 2013*), and it can create false memories for highly related associate words (*Deese, 1959*; *Roediger and McDermott, 1995*). In favor of the second (interference) account, pairing a single retrieval cue with multiple target responses could benefit from greater semantic differences between the targets, allowing for easier dissociation between them (*Underwood, 1969*). Indeed, increasing relatedness between tasks in some paradigms can increase interference (*Bower et al., 1994*; *McGeoch and McGeoch, 1937*; *McGeoch and McDonald, 1931*) and/or the rate of intruding material from one task to the other (*Postman, 1961*; *Osgood, 1946*; *Underwood, 1951*; *Dallett, 1962*; *Dallett, 1964*). Finally, in favor of the third (strengthening) account, there is evidence that we are not always passive during new learning: sometimes we 'think back' to, and thereby reactivate, prior experiences (*Hintzman, 2011*). These events, called recursive reminders, can occur when subjects are given explicit instructions or cues as reminders (*Chanales et al., 2019*; *Negley et al., 2018*; *Lustig et al., 2004*), or—more relevantly here—they can occur spontaneously when information is related (*Hintzman et al., 1975*; *Garlitch and Wahlheim, 2020*). Moreover, recursive reminders seem to create interdependence between old and new information, with preserved information about the temporal order of learning rather than source confusion and negative competition between the traces (*Hintzman, 2011*; *Wahlheim and Zacks, 2019*; *Jacoby et al., 2015*; *Ngo et al., 2021*). The recursive reminders account therefore predicts that semantic relatedness would promote RF and interdependence among memory traces. Altogether, the first account is clearly incorrect, but when and how strongly the countervailing forces of RI and RF from the latter accounts operate remains a central puzzle.

In examining these accounts more deeply, we will focus on a range of findings from experimental paradigms featuring associations between cues (A) and targets (B). In these paradigms, simply practicing the associations (i.e., seeing the cues paired with the same targets) ubiquitously (and obviously) produces RF of the original A-B association. One of the most studied deviations from this involves linking identical cues (A) with new targets (D) after A-B learning (*Briggs, 1954*; *Barnes and Underwood, 1959*). This paradigm is canonically referred to as A-B, A-D learning, and we will call it ΔTarget learning because it involves a change in the target. ΔTarget learning typically causes RI for the original A-B memory, likely due to competition between the target responses during retrieval (*Bower et al., 1994*; *Caplan et al., 2014*). However, this RI effect is known to be sensitive to the relationship between the old target (B) and new target (D), as RI generally decreases from substantial to near-absent when B and D are semantically related (*Osgood, 1946*; *Dallett, 1962*; *Barnes and Underwood, 1959*; *Osgood, 1948*; *Morgan and Underwood, 1950*; *Mehler and Miller, 1964*; *Kanungo, 1967*; *Young, 1955*; *Postman, 1964*; *Postman and Parker, 1970*). In other paradigms, interference can become reduced by encouraging subjects to integrate the two interfering pieces of information (*Anderson and McCulloch, 1999*; *Moeser, 1979*; *Carroll et al., 2007*; *Reder and Anderson, 1980*), suggesting that high semantic relatedness may reduce RI by making the related memories interdependent. These findings suggest that RI generally occurs when a cue is linked with competing target responses, but that increasing relatedness can reduce or overcome these effects, likely due to recursive reminders.

The idea is that competition between targets at retrieval causes RI makes a different prediction for lists with identical targets but new cues. Under such A-B, C-B learning conditions, which we call ΔCue learning, little to no RI occurs (*Twedt and Underwood, 1959*; *Keppel and Underwood, 1962*; *Houston, 1966*). In fact, when targets are identical and old and new cues are semantically related, RF occurs (*Hamilton, 1943*; *Bugelski and Cadwallader, 1956*). However, the level of relatedness may be more modest in this case, exposing a contrast between the role of cues and targets. Finally, changing both cues and targets at once, canonically referred to as A-B, C-D learning that we will

call ΔBoth learning, generally results in a completely different learning event (i.e., neither RI nor RF). Studies in which both cues and targets bear some level of relationship to the original A-B pair are scant. However, there have been cases where *either* the new cue was semantically related to the old cue but the targets were unrelated, the new target was semantically related to the old target but the cues were unrelated, or both new cues and targets shared some modest level of relatedness with the old ones; in each of these cases, RI for the original A-B association has been observed when testing occurred after short retention intervals (on the order of minutes) (*McGeoch and McGeoch, 1937*; *Bugelski and Cadwallader, 1956*; *Baddeley and Dale, 1966*; *Saltz and Hamilton, 1967*). However, to our knowledge, no study has investigated longer-term memory in cases where the new cues and targets were both highly related to the old ones.

To conceptualize this complex array of RI/RF effects, Osgood proposed three continuous directions along which relatedness influences memory (*Eich, 1982*; *Mensink and Raaijmakers, 1988*). First, when cue identity is held constant, he reasoned that there must be some point along the ΔTarget line between an unrelated target and an identical target (i.e., from A-B, A-D to A-B, A-B) at which RI shifts to RF. Second, holding the unrelated target constant and manipulating the cue from an identical

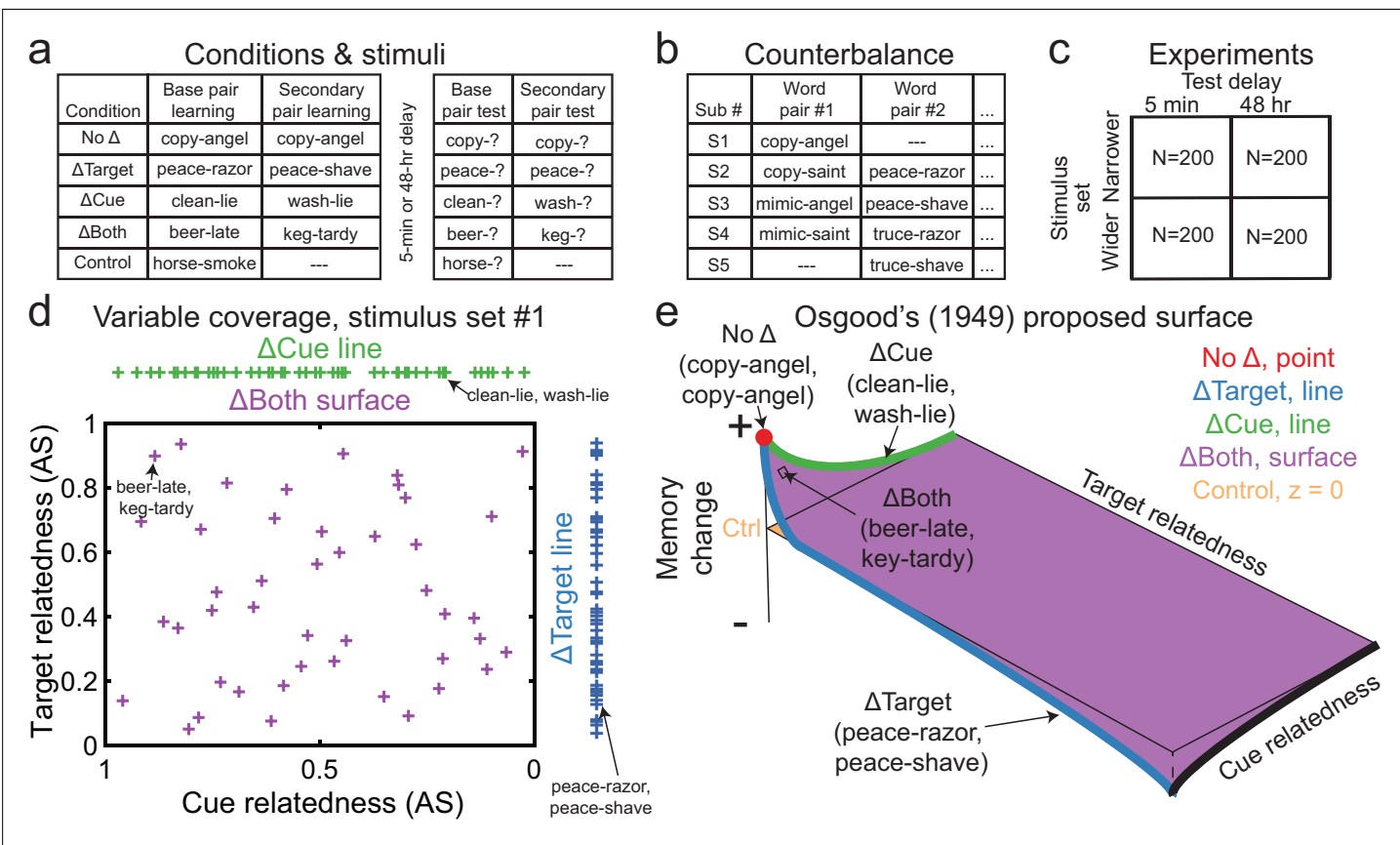

**Figure 1.** Overview of conditions, stimuli, experiments, variables, and Osgood's predictions. (**a**) After base pair learning, pairs were divided into five experimental conditions for secondary pair learning. After a 5-min or 48-hr delay, both base and secondary pairs were tested. (**b**) Word pairs were counterbalanced every five subjects into conditions. (**c**) Overview of the first four experiments by delay and stimulus set. (**d**) Coverage of variables across associative strength (AS) values in the stimulus set with a narrower range of semantic relationships. Crosses along the ΔCue (green) and ΔTarget (blue) lines show distributions of cue and target relatedness, respectively. Purple crosses inside the surface (scatterplot) show the distribution of bivariate cue and target relatedness in the ΔBoth condition. (**e**) Experimental data hypothetically conforming to *Osgood, 1949* proposed surface. Cue and target relatedness span the y- and x-axes, respectively, while memory change for each condition spans the z-axis, relative to the control condition on the z=0 surface. In (**d**) and (**e**), example word pairs from (**a**) were labeled for illustrative purposes. The x-axes were reversed from normal convention to correspond to Osgood's surface. See also *Figure 1—figure supplement 1* for visualizations using the stimulus set with wider semantic relatedness and *Supplementary files 1-2* for all stimuli.

The online version of this article includes the following figure supplement(s) for figure 1:

**Figure supplement 1.** Wider stimulus set examples and coverage.

cue to an unrelated cue (i.e., from A-B, A-D to A-B, C-D), he reasoned that RI should disappear: the pair becomes an entirely new relation. Third, when the target is identical, Osgood noted that RI generally does not occur and drew a ΔCue line from unrelated, which produced no memory change, to maximum RF at identity (i.e., from A-B, C-B to A-B, A-B). From these three predictions, Osgood interpolated a full three-dimensional surface depicting how changes in cues, targets, or both should be expected to affect memory performance (*Figure 1*).

Despite the foundational nature of relatedness for learning and memory, the full space of Osgood's predictions and the conditions under which RI versus RF occurs under various paired-associate learning arrangements has remained largely untested. To the extent that prior studies have examined subregions of this space, they have relied upon differing methodological approaches, making it challenging to compare findings within a unified framework. Adequately testing these predictions requires a suite of associative learning studies that estimate retroactive and interdependence effects across a wide range of stimuli varying the semantic relatedness of cues, targets, or both. Such a study presents two primary challenges. First, it requires the ability to obtain reliable measures of semantic relatedness for arbitrary pairs of stimuli, which have become increasingly accessible given rapid progress in models of distributional semantics and large-scale data sets collecting associative norms. Second, obtaining adequate coverage over the full relatedness space requires a lot of data: each point in this space must be estimated from measurements of memory performance across multiple participants.

We addressed these challenges across five large experiments (N=1000). Subjects were initially given a list of 45 unrelated word pairs (e.g., sick-push), which we will call *base pairs*. Later, during the learning of what we will call *secondary pairs*, we included four within-participant experimental conditions: pairs in the No Δ condition appeared unchanged (sick-push), while other pairs appeared in the ΔTarget (sick-shove), ΔCue (ill-push), or ΔBoth (ill-shove) conditions. A final subset of pairs in a control condition remained unshown in this phase (*Figure 1A*). Subjects were trained until each pair was correctly retrieved once, after which the pair dropped out from later rounds of learning. Condition assignments were counterbalanced, such that the same base pair was rotated across these five secondary pair conditions across every five subjects (*Figure 1B*). In our initial experiments, we used a stimulus set with a narrow range of relatedness values, corresponding to the direct associative pair strength. In later experiments, to address how these initial results generalized beyond local semantic neighborhoods of direct associations, we used a stimulus set with a wider range of relatedness that included truly unrelated associations. Additionally, interference often differs depending on the delay between learning interfering material and test (*Lustig et al., 2004*; *Chan, 2009*; *Baran et al., 2010*; *Ortega et al., 2015*; *Liu and Ranganath, 2019*; *Wixted, 2004*), and we therefore fully crossed the narrower and wider stimulus sets with two different test delays occurring 5 min and 48 hr after secondary pair learning (*Figure 1C*). Finally, it has been extensively found that engaging in retrieval strongly benefits long-term memory relative to only restudying information (*Roediger and Karpicke, 2006a*; *Antony et al., 2017*; *Carpenter and Yeung, 2017*). We therefore conducted a final experiment exploring whether the results changed if subjects only studied during learning (while controlling for overall exposure to the pairs).

Our large sample sizes allowed us to test the memorability of each base pair in each condition across subjects, eliminating the incidental effects of the individual base pairs. We measured retroactive effects using each of the following conditions as a difference from the control condition: No Δ at the cue and identity point, ΔTarget as a line of values at cue identity spanning target relatedness, ΔCue as a line of values at target identity spanning cue relatedness, and ΔBoth as a surface spanning bivariate cue and target relatedness (see *Figure 1D* for illustration using the narrower stimulus set and *Figure 1—figure supplement 1* using the wider set). Putting these different conditions together, we show how our paradigm could produce Osgood's proposed surface in *Figure 1E*, which we test empirically below. Note that if increasing relatedness among word pairs along one or more dimensions increased RI, it would run contrary to Osgood's predictions. Conversely, if increasing relatedness increased RF, it would support his predictions. Such results would also support recursive reminder theory (*Jacoby et al., 2015*), which we believe offers a mechanistic explanation of Osgood's proposed surface because it predicts that retroactive benefits increase as reminders become more likely (such as with greater semantic relatedness). A further prediction of this theory is that relatedness would promote interdependence between associated memory traces.

## Results

### High semantic relatedness produced profound retroactive facilitation, especially at long delays

We began by establishing the pattern of retroactive memory effects across our five main conditions in a regime of high overall semantic relatedness. In our first experiment, we operationalized semantic relatedness in terms of associative strength (AS) (*Nelson et al., 1998a*), the (empirical) probability that a second word is freely generated as a response to a given word, as estimated from a large independent population. We chose words for the secondary pair learning phase that predicted their corresponding base words with AS values that were quasi-evenly spaced from the lowest values (pious→holy, which was only generated with probability 0.03) to the highest (moo→cow, which was only generated with probability 0.96). We also imposed a 48-hr delay before testing. We measured base pair memory performance, or accuracy in providing the correct target word at test, across conditions using a one-way (No Δ, ΔTarget, ΔCue, ΔBoth, and control) ANOVA, collapsing across all levels of semantic relatedness. We found that condition significantly affected overall memory [$F(3.9,775.3)=126.2$, $p<0.001$]. Follow-up t-tests indicated differences across all pairwise conditions, following No Δ>ΔCue>ΔTarget>ΔBoth>control (all adjusted $p<0.002$), which can be found in the top right of *Figure 2*. In other words, related associates generally produced RF, although holding the target constant (i.e., ΔCue) benefitted memory more than holding the cue constant (i.e., ΔTarget). Critically, we also found significant RF in the ΔBoth condition, even though there were no overlapping words between the initial and later-learned pairs.

One possible explanation for the lack of RI effects is the relatively long (48 hr) delay before the final test. Many forms of interference are known to depend on delay (*Chan, 2009*; *Baran et al., 2010*; *Ortega et al., 2015*; *Liu and Ranganath, 2019*) (see also *Jonker et al., 2018*) and RI effects in particular typically decrease with delay as the interfering material becomes forgotten (*Lustig et al., 2004*; *Wixted, 2004*). We therefore reasoned that we may find RI if we repeated the experiment with a shorter delay of only 5 min. In this experiment, we again found a significant difference in performance across condition [$F(3.9,780.6)=37.8$, $p<0.001$] (*Figure 2*, top left) but with no evidence of RI. Pairwise t-tests indicated performance followed No Δ>ΔCue>ΔTarget>ΔBoth=control (all adjusted $p<0.014$; '=' indicates $p=0.27$), consistent with our findings for the longer delay. Taken together, these two experiments demonstrated that high semantic relatedness between initial and later-learned information produced RF.

### Under a wider range of semantic relatedness, condition and delay determined retroactive effects

The prior results showed high RF for nearly all experimental conditions. These results were especially surprising in the ΔTarget condition, as RI is ubiquitous in these paradigms, especially after short delays. The primary deviation in our variant was that cue and target associations were strongly related. We therefore considered that these effects arose because even the least-related cue and target associations (e.g., pious→holy) were highly similar in the overall semantic space of words because they were all identified from a local semantic neighborhood: all words used in the secondary phase were produced in a single step of free association from the base word. To test this possibility, we expanded the distribution of relatedness to pairs that spanned the full range of semantic relatedness. To quantify relatedness in this stimulus set, we used the cosine similarity [$\cos(\theta)$] between GloVe vector embeddings. These high-dimensional semantic representations were trained on word-word co-occurrence in large text corpora and strongly align with human similarity judgments (*Pennington et al., 2014*). We chose GloVe values distributed quasi-evenly from –0.14 to 0.95, which encompassed a wide range of associations from those that would be considered unrelated (e.g., sap→laugh) to those which appear as one-step semantic relationships according to our earlier measure of association strength (e.g., blue→red). Aside from the wider stimulus set, the learning procedure was identical.

We tested these new stimuli under both long and short delay conditions for comparison with our earlier results. In the 48-hr delay experiment, we again found that base pair memory differed strongly across conditions [$F(4,796)=128.3$, $p<0.001$], with pairwise t-tests indicating that No Δ>ΔCue>ΔTarget>control=ΔBoth (all adjusted $p<0.002$; '=' indicates $p=0.056$), again showing facilitation overall (*Figure 2*, lower right). In the 5-min delay experiment, however, the results were

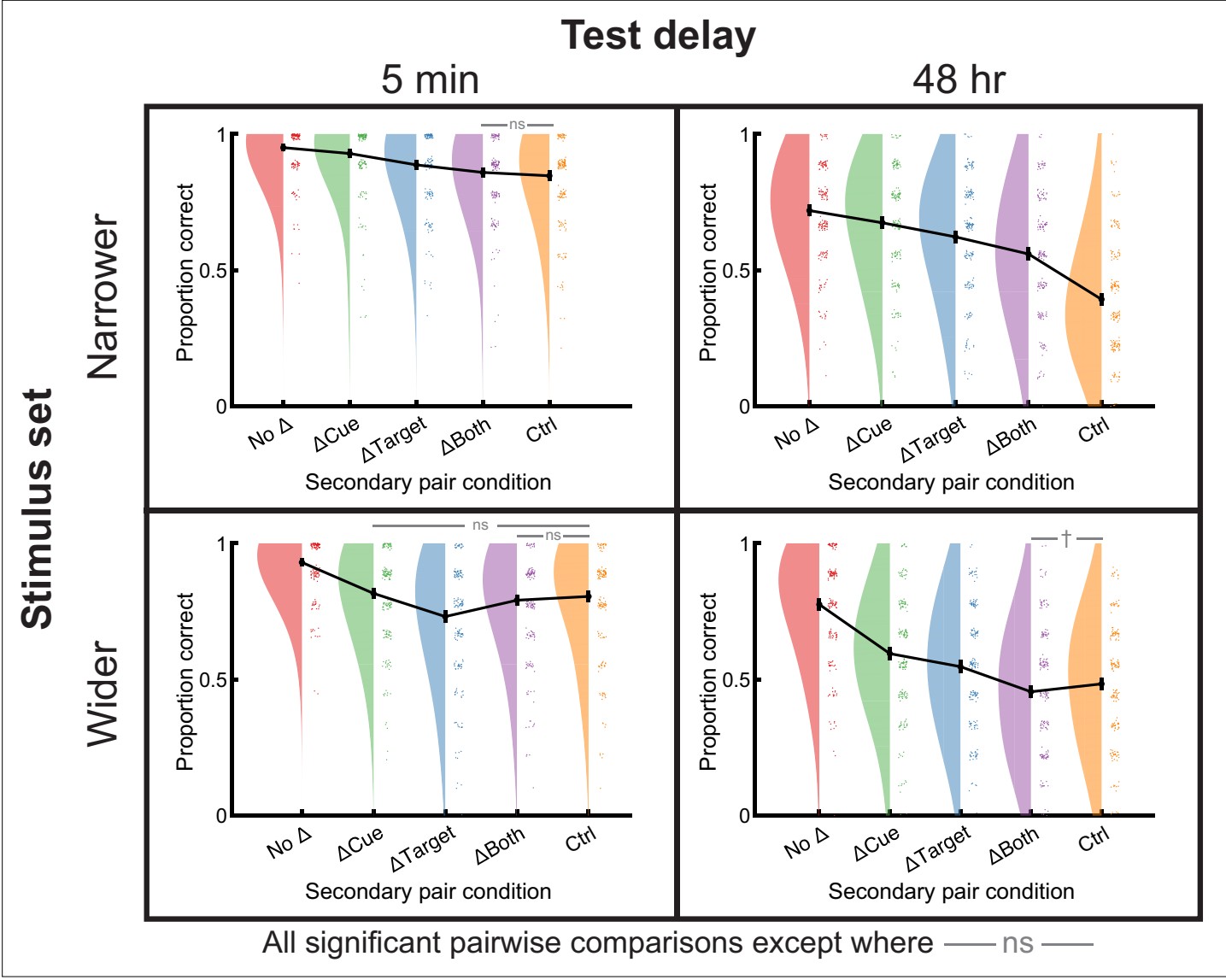

**Figure 2.** RF versus RI differed by overall stimulus set relatedness, delay, and word pair condition. The narrower stimulus set (top row) featured only single-step semantic associations between base and secondary cues and targets, whereas the wider stimulus set (bottom row) featured a full range of semantic relationships. All comparisons were significant except those labeled with gray bars and 'ns' (p>0.1) or † (0.05<p<0.1). Data points from individual subjects were jittered slightly for better visualization. See also *Figure 2—figure supplement 1* for secondary pair memory and *Supplementary file 3* for numerical results. RF, retroactive facilitation; RI, retroactive interference.

The online version of this article includes the following figure supplement(s) for figure 2:

**Figure supplement 1.** Secondary pair memory changes depending on overall stimulus set relatedness, delay, and word pair condition.

strikingly different. Base pair memory differed across conditions [$F$(3.7,742.4)=59.7, p<0.001], but we found evidence of RI in the ΔTarget condition, such that No Δ>ΔCue=control=ΔBoth>ΔTarget (ΔCue vs. control p=0.38; control vs. ΔBoth p=0.29; ΔCue and ΔBoth did differ, p=0.04; all others, p<0.001) (*Figure 2*, lower left). We therefore successfully replicated the classical RI effects, but only under the conditions of low average relatedness and a short delay. Additionally, the finding that the ΔBoth condition did not significantly differ from control in both experiments suggests that subjects may mentally categorize these pairs as novel pairs (resembling classical C-D pairs) when overall relatedness was low. For results from secondary pair testing in all experiments, please see *Figure 2—figure supplement 1*.

## Target relatedness produced RF and scaffolded new target learning

Having established condition-level effects of facilitation consistent with Osgood's predictions (*Osgood, 1949*), we next conducted a more direct test by predicting facilitation as a function of relatedness at the level of word pairs. In the first of these analyses, we focused on word pairs in the ΔTarget condition. We subtracted the proportion of subjects successfully recalling each pair in the control condition from the proportion in the ΔTarget condition, yielding a measure for each individual word pair that is positive for evidence of RF and negative for evidence of RI. We then performed linear regression analyses between this retroactive measure and the semantic relatedness of the pair, using AS or GloVe values depending on the stimulus set. These analyses allowed us to ask whether word pair memorability was directly correlated with relatedness at the item level.

We found that higher semantic relatedness between targets produced greater facilitation in all experiments (all $p<0.05$; *Figure 3A*) except for the narrower stimulus set, 5-min experiment, where we found near-ceiling memory performance ($p=0.75$). Intriguingly, the results from the 5-min and 48-hr delay experiments with the wider stimulus set further clarified how both delay and semantic relatedness additively determined RI or RF. In the 5-min experiment, we found RI for pairs with very low relatedness, which would be conventionally categorized as 'unrelated,' as indicated by the significantly negative y-intercept in the regression (lower left of *Figure 3A*). As relatedness increased, however, items entered a region that did not differ from the control condition. In the 48-hr experiment, there was no difference from the control condition for pairs with very low relatedness, as indicated by the insignificant y-intercept in the regression, but with increasing relatedness, we found significant RF (lower right of *Figure 3A*). Additionally, in the experiment using the narrower stimulus set and shorter 5-min delay, the ΔTarget condition still produced facilitation compared to the control, suggesting that with high enough overall relatedness, temporary RI effects can fully cross over into RF (upper left of *Figure 3A*).

The recursive reminders account predicts that when one retrieves an initial pair during new learning, the two become interdependent. It also predicts that reminders should increase with relatedness. Therefore, we next asked whether relatedness promoted interdependence between initial and later-learned pairs. We defined interdependence as the proportion of base pair target-secondary pair target duos that were both correct or both incorrect across subjects. For example, if subjects tended to recall 'peace-razor' during base pair testing and 'peace-shave' during secondary pair testing, or failed to recall both of them, these would be interdependent, whereas if only one of the two memories was recalled as often as both or neither of the words, these would be independent. We then correlated this interdependence measure with semantic relatedness. We found memory dependence increased with higher relatedness ($p<0.01$), except in the narrower stimulus set, 5-min experiment that previously showed near-ceiling memory performance (*Figure 3B*). Therefore, target relatedness simultaneously resulted in strengthened base pairs and enhanced interdependence between base and secondary pairs.

We also asked whether target relatedness would increase intrusions, or errors from the secondary pair list into the base pair list. That is, we wanted to contrast two accounts. Under one account, the targets may merge or compete, leading to confusion about the list contexts (e.g., peace-razorshave). Theoretically, this account could produce some intrusions in addition to RF; indeed, lack of interference in RI studies wherein targets are related has been posited to stem from a 'loss of differentiation' between semantically related sources (*Postman, 1961*), and other studies have found greater intrusion errors with increasing relatedness (*Osgood, 1946*; *Underwood, 1951*; *Dallett, 1962*; *Dallett, 1964*). Under the recursive reminders account, highly related new targets would simultaneously strengthen old memories due to reminders of the base pair list and would be scaffolded to the cue as part of the secondary pair list, meaning the list contexts remained interdependent, yet distinguishable (e.g., peace-razor-base list/peace-shave-secondary list). We therefore asked whether relatedness increased across-list intrusions of the new target response into the base pair list by correlating the across-subject intrusion rate with target relatedness. In fact, intrusions significantly decreased in the wider stimulus set, 5-min experiment ($p<0.001$) and otherwise did not increase with target relatedness in any experiment (all $p>0.08$; *Figure 3—figure supplement 1*), supporting the recursive reminders account.

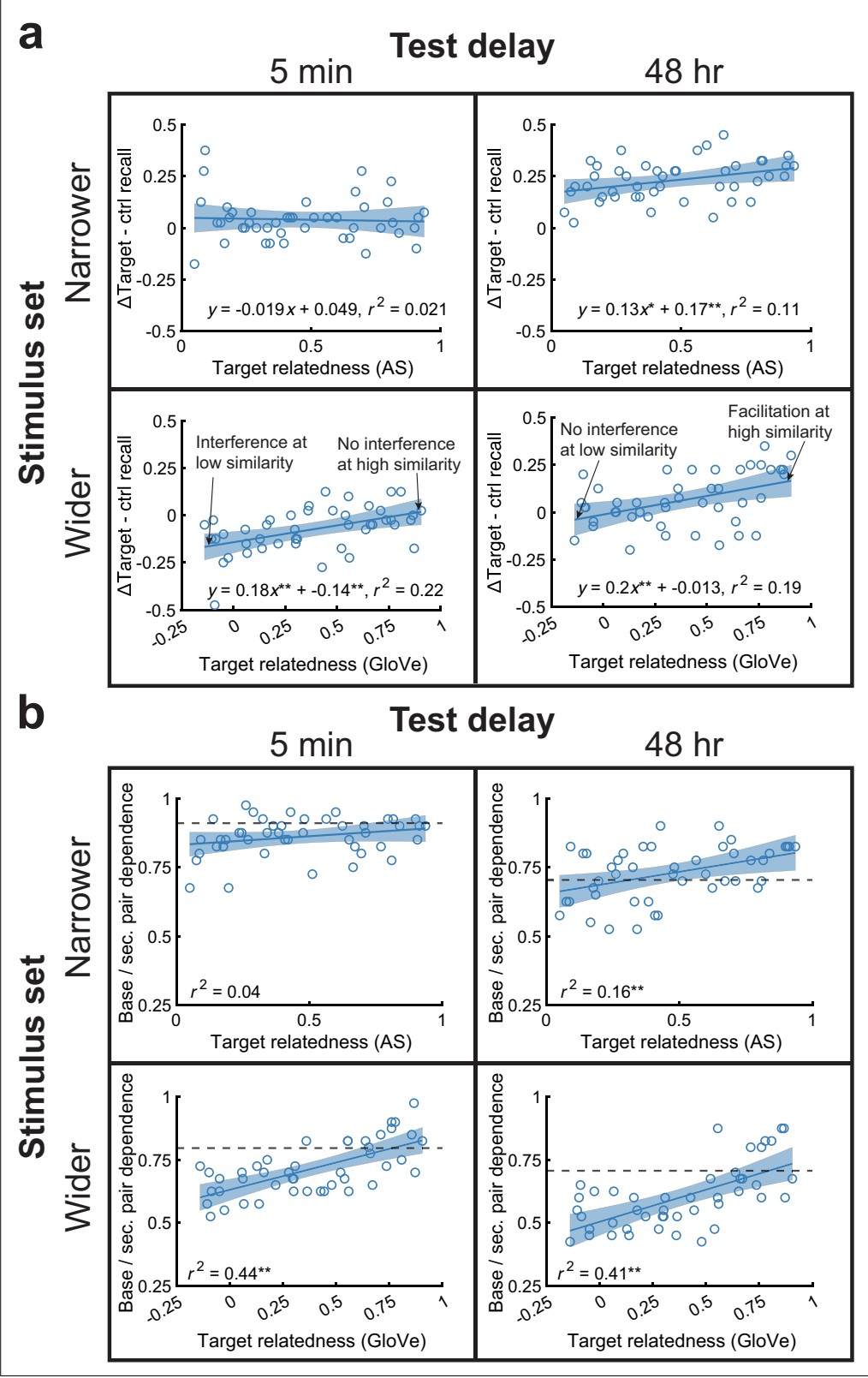

**Figure 3.** Target relatedness retroactively benefited memory and created interdependence between base and secondary pairs. (**a**) Across-subject memorability for each base pair was plotted against the target semantic relatedness, with AS and GloVe values in the top and bottom rows, respectively. Relatedness improved memory in all experiments except in the narrower stimulus set, 5-min delay experiment, where overall memory approached

*Figure 3 continued on next page*

*Figure 3 continued*

ceiling performance. RI occurred with especially low relatedness in the wider stimulus set, 5-min delay experiment, but this interference disappeared with high relatedness. In the wider stimulus set, 48-hr delay experiment, we found no interference with low relatedness and facilitation with high relatedness. (**b**) Within each base pair target-secondary pair target duo, we plotted across-subject memory dependence against semantic relatedness in all experiments. Higher correlation values indicate that subjects tended to remember or forget both targets in the duo together. Relatedness increased these correlations in three experiments, excluding the narrower stimulus set, 5-min delay experiment. Thick dotted lines show the 95th percentile threshold of dependence levels against all other pairs. See also *Figure 3—figure supplement 1* for intrusion data from this condition. RF, retroactive facilitation; RI, retroactive interference.

The online version of this article includes the following figure supplement(s) for figure 3:

**Figure supplement 1.** Intrusions did not increase with target relatedness in the ΔTarget condition.

## Cue relatedness had no significant linear effect on base pair memory

Overall, performance in the ΔCue condition showed consistent RF across experiments. We next asked whether memorability differed as a function of cue relatedness, as it did for target relatedness. We found no significant relationship between cue relatedness and word memorability in the ΔCue – control condition in any experiment, though there were marginally significant effects in the narrower stimulus set, 5-min delay ($p=0.085$) and wider stimulus set, 5-min delay experiments ($p=0.056$) (others, $p>0.28$; *Figure 4*). We also repeated the memory dependence analyses between base pair target-secondary pair target duos in the ΔCue condition. We found inconsistent results: dependence increased with cue relatedness in the narrower stimulus set, 5-min experiment ($p=0.01$) and the wider stimulus set, 48-hr experiment ($p<0.001$), but not in the others (both $p>0.23$; *Figure 4—figure supplement 1*). Therefore, even though word pairs in the ΔCue condition showed RF overall, the relationship between cue relatedness and memorability was non-existent to weak, standing in contrast to those in the ΔTarget condition. Memory dependence between base and secondary pairs appeared more statistically reliable, though it was unclear under which conditions dependence between base and secondary pair memories arose.

## Bivariate cue and target relatedness improved long-term memory and increased memory dependence

Overall, we found that performance in the ΔBoth condition showed RF in the narrower stimulus set, 48-hr delay experiment, but it did not differ from the control condition in the other experiments. We next asked whether these retroactive effects differed as bivariate values of cue and target relatedness in all experiments, with particular interest in the narrower stimulus set, 48-hr delay experiment. To do this, we computed base pair memorability in the ΔBoth and control conditions for each word pair, and then we computed locally smoothed surfaces based on memorability at each bivariate cue and target relatedness value (see Materials and methods). To assess statistical significance, we first looked for clusters of values whereby the true values exceeded those expected by chance by randomly scrambling the conditions. We then used 1000 permutation tests to assess the size of clusters exceeding this threshold that we might expect due to chance, and we asked whether any observed true clusters exceeded this threshold. We found a significant cluster with high levels of both cue and target relatedness in the narrower stimulus set, 48-hr delay experiment ($p=0.001$) and not the others (all $p>0.41$), demonstrating that long-term RF occurs with high bivariate levels of relatedness (*Figure 5A*). We also computed memory dependence between base pair target-secondary pair target duos in the ΔBoth condition, and we similarly created locally smoothed surface plots of these dependence values. Like the memorability analyses, we found a cluster at high levels of both cue and target relatedness only in the narrower stimulus set, 48-hr delay experiment (*Figure 5B*). Therefore, results from the narrower stimulus set, 48-hr delay experiment concur with findings in the ΔTarget condition whereby relatedness simultaneously strengthens base pair memory and increases dependence between base and secondary pair memory. For qualitatively similar results correlating retroactive benefits and memory dependence, except with a linear measure against the added value of cue+target relatedness, see *Figure 5—figure supplement 1*.

We next explored whether cue or target relatedness differentially affected memorability and base-secondary pair dependence within the ΔBoth condition. We found that target relatedness correlated

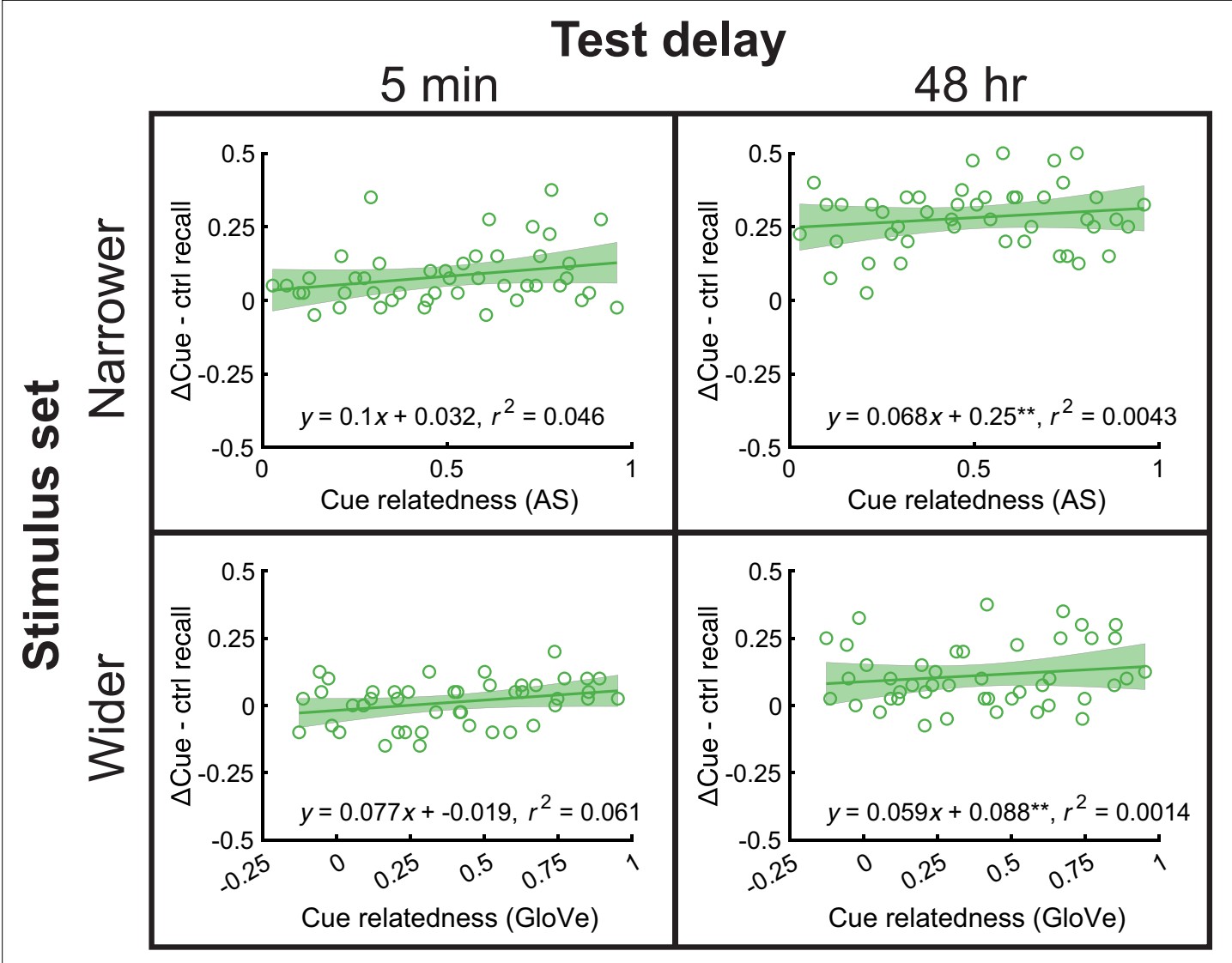

**Figure 4.** Cue semantic relatedness has no consistent retroactive effect. Across-subject memorability for each base pair – control was plotted against cue relatedness (top: AS; bottom: GloVe). Relatedness had no effect on memory in any condition. See also *Figure 4—figure supplement 1* for dependence correlations.

The online version of this article includes the following figure supplement(s) for figure 4:

**Figure supplement 1.** Cue relatedness showed a mixed relationship with memory dependence in the ΔCue condition.

with ΔBoth memorability (r=0.38, p=0.01), whereas cue relatedness did not (r=0.17, p=0.26). Furthermore, the target relatedness correlation survived significance when performing partial correlations controlling for cue relatedness (r=0.39, p=0.009). Conversely, we found that cue relatedness correlated with base-secondary pair dependence in the ΔBoth condition (r=0.30, p=0.04), whereas target relatedness did not (r=0.12, p=0.42), and the cue relatedness correlation survived significance when performing partial correlations controlling for target relatedness (r=0.31, p=0.04). Therefore, although our primary analyses in the ΔBoth condition focused on the bivariate effects of cue and target relatedness, the two measures have dissociable impacts on memorability and dependence.

## Osgood-style retroactive and dependence surfaces

What happens to an association after its initial formation, and when do two memories become linked? We now attempt to answer these questions by consolidating all experimental conditions in the style of Osgood's surfaces (*Osgood, 1949*). Surfaces from all retroactive memory results can be viewed

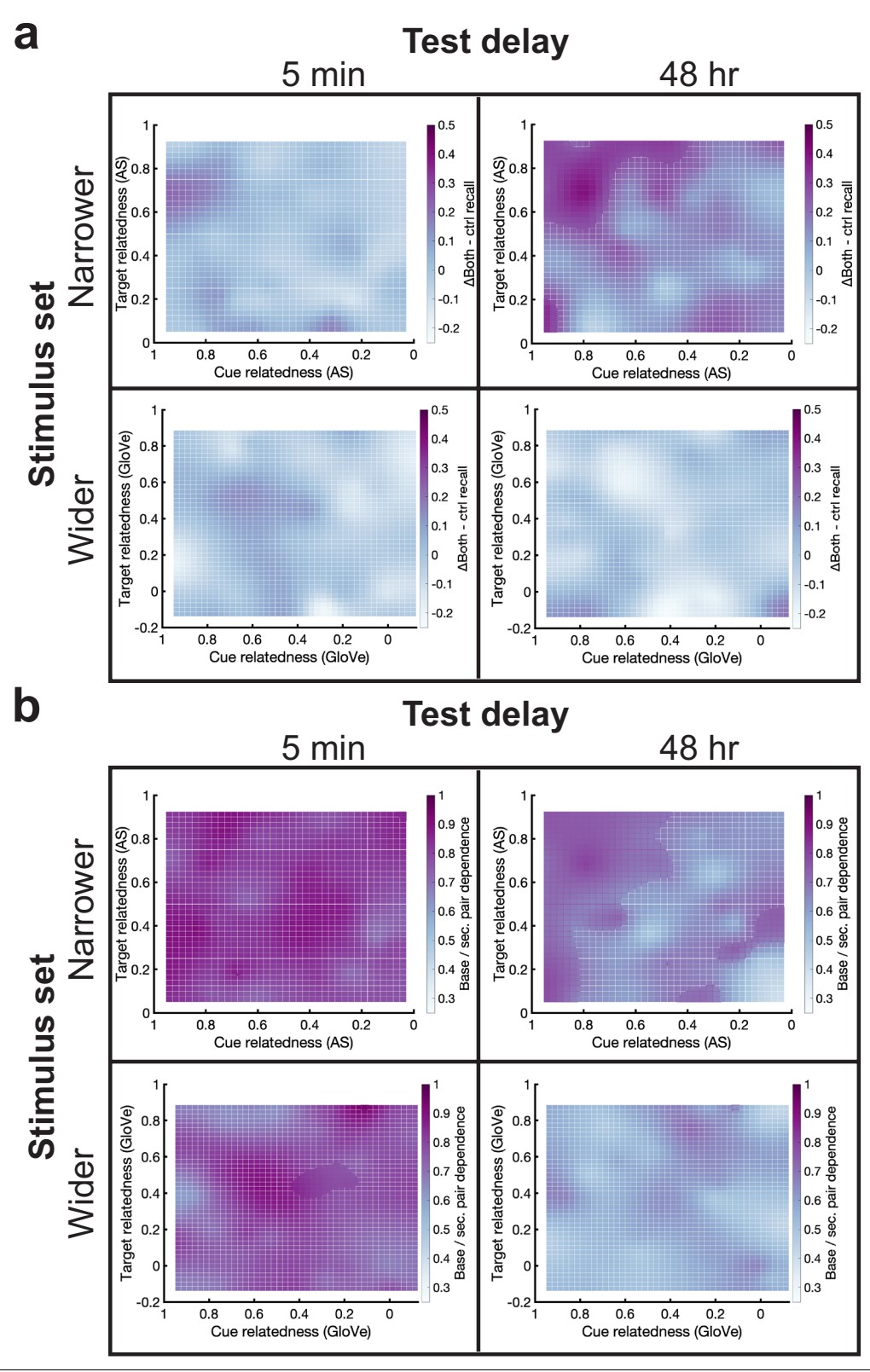

**Figure 5.** High cue and target relatedness promoted long-term RF and increased interdependence. (**a**) We plotted a smoothed surface of ΔBoth – control memorability values against cue and target relatedness on the *x*- and *y*-axes (top: AS; bottom: GloVe). Under the narrower stimulus set in the 48-hr delay experiment, memorability increased at high levels of both cue and target relatedness. (**b**) Similar to (**a**) except depicting smoothed surfaces of base-

*Figure 5 continued on next page*

*Figure 5 continued*

secondary pair dependence. High values of cue and target relatedness increased dependence in the narrower stimulus set, 48-hr delay experiment. Sections of these plots with purple grid marks were significant, whereas those with white marks were not. See *Figure 5—figure supplement 1* for linear contrasts between cue+target relatedness and memory and dependence measures. RF, retroactive facilitation.

The online version of this article includes the following figure supplement(s) for figure 5:

**Figure supplement 1.** High cue+target relatedness promotes long-term RF and increases interdependence.

together in *Figure 6*, though we will start by examining the narrower stimulus set, 48-hr experiment (right side of *Figure 6A* and upper right of *Figure 6—video 1*). Consider that after base pair learning, the strength of any given association sits along the x-y plane, where target and cue relatedness of a putative secondary pair lie along the x- and y-axes, respectively. If no related pairs occur during secondary pair learning (control condition), it remains along this axis (orange). If during secondary pair learning, the same pair is learned again (No Δ condition), it sits at the target identity, cue identity point (red). If cues remain and targets change (ΔTarget condition), it varies by target relatedness along the cue identity line (blue). If targets remain and cues change (ΔCue condition), it varies by cue relatedness along the target identity line (green). Finally, if both cues and targets change (ΔBoth condition), it rests upon the surface as a bivariate function of cue and target relatedness (purple). Examining results from this experiment, as secondary pair relatedness approaches or reaches full identity along multiple dimensions, memorability improves (though note that the linear relationship along the target identity line is not significant). All conditions from all other experiments are shown on the left of *Figure 6A* and the other quadrants of *Figure 6—video 1*, which shows rotations around the 3-D space.

In addition to considering retroactive base pair effects in isolation, we similarly plotted memory dependence between base and secondary pairs as a function of cue and target relatedness. Examining the narrower stimulus set, 48-hr experiment (right of *Figure 6B* and upper right of *Figure 6—video 2*), dependence increased with relatedness along multiple dimensions, including near the cue identity, target identity portion of the bivariate surface, in a manner resembling the retroactive effects. All conditions from all other experiments are shown on the left of *Figure 6B* and other quadrants of *Figure 6—video 2*; once again, the lack of dependence along the ΔBoth surface in the wider stimulus set experiments accords with a likely independence between old and new pairs under lower average relatedness. Overall, these results strikingly show how semantic relatedness—examined via multiple types of associations—produced retroactive benefits and memory dependence.

## Examining retroactive memorability and memory dependence effects with a common metric and with other relatedness metrics

Above, we featured the relatedness dimensions we originally chose to continuously span the stimulus spaces (AS for the narrower stimulus set and GloVe for the wider stimulus set). However, we wanted to address two remaining points. First, the two stimulus sets span variable ranges of relatedness. We kept analyses for these experiments separate because sensitivity to the overall distribution of relatedness within a particular session could affect subject performance; nevertheless, we acknowledge that using different ranges could result in the effects disappearing when the data become combined under a unified metric. To address this concern, we combined across-subject memorability and dependence across stimulus sets in each experimental condition within a particular test delay (e.g., the narrower and wider stimulus sets within the 5-min delay experiments). Next, we correlated these with measures with GloVe values (*Figure 6—figure supplement 1*). Notably, none of the prior significant effects disappeared under this analysis. Rather, retroactive and memory dependence effects in the ΔCue condition actually became significant under this common metric (likely due to increased power), yet they remained weaker than in the ΔTarget condition, consistent with our prior results.

The second remaining point is that many other relatedness metrics exist (besides AS and GloVe), which can be broadly categorized into 'internal' models relying on relationships within associative semantic networks and 'external,' vector-based models based on recently developed algorithms trained on large amounts of text that can measure word-word relationships (*De Deyne et al., 2017*). In some cases, internal models outperform external models at capturing paired-associate memory effects (*Steyvers et al., 2005*). Additionally, semantic network relationships can predict paired-associate

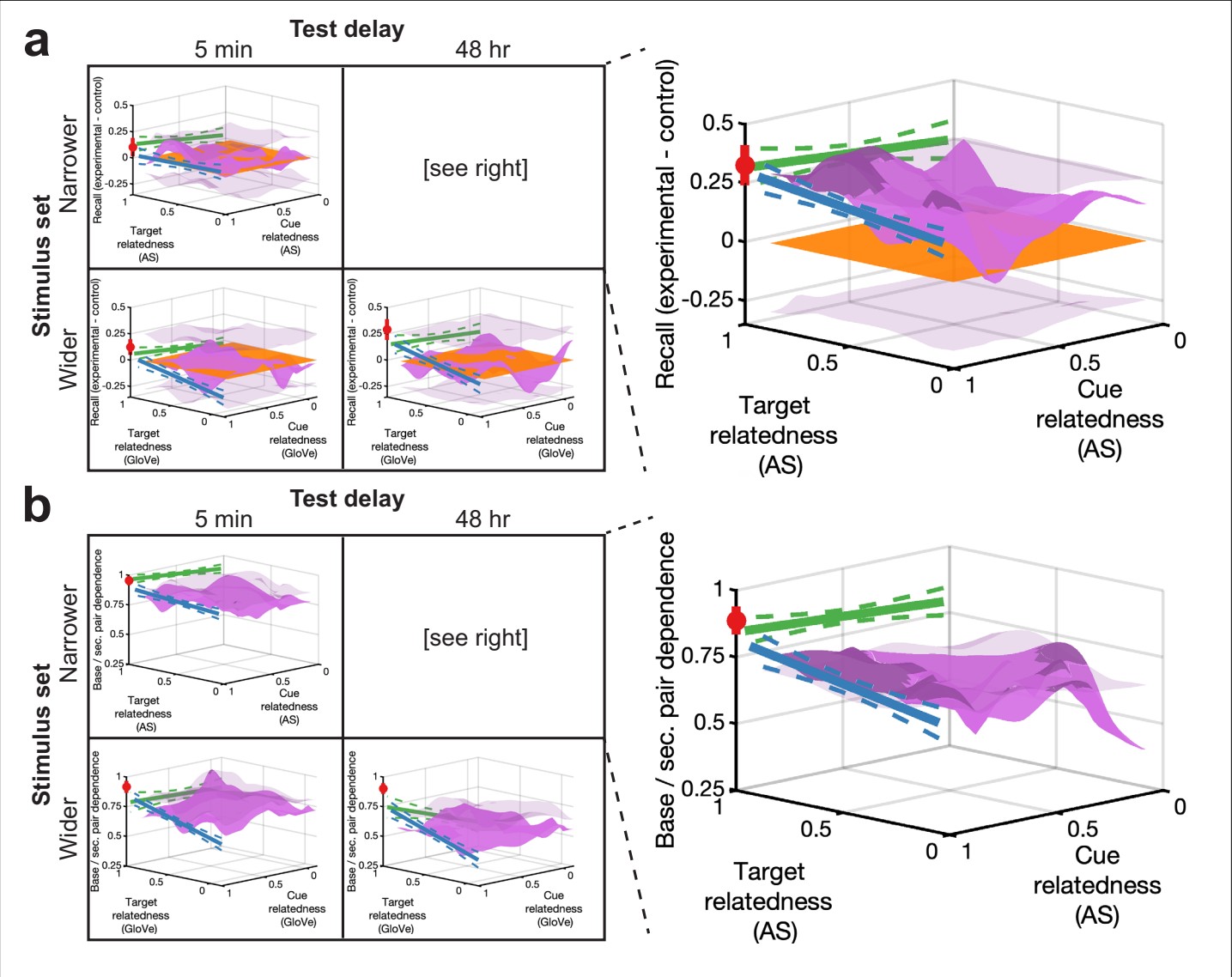

**Figure 6.** Osgood-style surfaces depicting retroactive effects and dependence. (**a**) We plotted all conditions (vs. control) from all experiments in three-dimensional coordinates, with cue and target relatedness on the y- and x-axes, respectively, and retroactive memory change on the z-axis, with RF and RI in the positive and negative directions, respectively. ((**a**), right) For the narrower stimulus set, 48-hr delay experiment, we plotted memory for the No Δ – control condition (±across-pair standard deviation) at the cue identity, target identity corner point (red circle). We plotted ΔTarget – control condition memory along the cue identity line against target relatedness (± standard error from the ordinary-least-squares regression fit) (blue), and we plotted ΔCue – control condition memory along the target identity line against cue relatedness (± standard error from the ordinary-least-squares regression fit) (green). We plotted ΔBoth – control condition memory as a locally smoothed surface as a bivariate function of cue and target relatedness (purple). Transparent surface grids above and below zero represent p<0.01 significance boundaries from permutation tests, beyond which the surface is significant, as indicated by a darker shade of purple. (left) Similar plots created for all conditions from the other experiments. (**b**) Dependence for all experiments and conditions formatted similarly to (**a**). RF, retroactive facilitation; RI, retroactive interference.

The online version of this article includes the following video and figure supplement(s) for figure 6:

**Figure supplement 1.** Semantic relatedness benefits memory and dependence when combining data sets under a common GloVe metric.

**Figure 6—video 1.** Rotations of Osgood-style retroactive memory surfaces from Figure 6.
https://elifesciences.org/articles/72519/figures#fig6video1

**Figure 6—video 2.** Rotations of Osgood-style memory dependence surfaces from Figure 6.
https://elifesciences.org/articles/72519/figures#fig6video2

memory beyond single steps to nearby neighbors, with significant benefits shown up to two (*Nelson et al., 1997*; *Nelson and Zhang, 2000*) or three (*Kenett et al., 2017*) semantic steps. We therefore included the following relatedness factors based on semantic networks: forward AS, or cue→Δcue and target→Δtarget AS rather than the backward (e.g., cue←Δcue) AS measure used in our analyses above; backward mediator strength, which calculates the cumulative strength of all secondary associations (e.g., mane-tiger via the mediator, lion, or cue←[mediator]←Δcue) and has been shown to predict memory independently from direct AS (*Nelson and Zhang, 2000*; *Nelson et al., 2003*); weighted path length, whereby we find the shortest path in a semantic network generated by free association norms and add up their summed weights between each node; and spreading activation strength, wherein we start from the target word and follow all edges to nearby nodes (words) up to three steps and add up activation values weighted by their association norms (see *De Deyne et al., 2017*; *Hills et al., 2015*; *De Deyne et al., 2016* for similar approaches). In addition to GloVe, we also used the following external models: word2vec cos(θ), wherein words are represented by vectorized representations based on training a neural network on a large text corpus (*Mikolov et al., 2013*); and latent semantic analysis (LSA) cos(θ), which captures contextual similarity between words/documents via projections into a high-dimensional semantic vector space (*Landauer and Dumais, 1997*). See *Supplementary files 4-5* for correlations among these metrics for our stimuli and *Supplementary file 6* for relationships with base pair memory and memory dependence. Additionally, since relatedness generally affected memorability and dependence, see *Supplementary file 7* for direct correlations between memorability and dependence and *Supplementary file 8* for correlations between relatedness and both memorability and dependence separately while controlling for the other measure. Although the results differed somewhat by experiment and condition, the backward AS and GloVe measures we used in our analyses above captured the same general effects.

## When subjects learned by studying, relatedness retroactively benefited memory in the ΔTarget condition but did not increase dependence

The prior experiments required one successful retrieval per word pair during learning. Prior research suggests that retrieval produces profound long-term memory benefits relative to a different learning strategy of studying (e.g., *Roediger and Karpicke, 2006b*) and may also differ in ways relevant to our effects. For instance, subjects may engage in more mental elaboration during retrieval than study, which may help form semantic mediators that can aid in retrieving a memory trace (e.g., retrieving 'mother-child' may activate the mediator, 'father') (*Carpenter and Yeung, 2017*) and can have benefits for related material (*Chan et al., 2006*). Therefore, our final experiment used the narrower stimulus set and a 48-hr delay, but subjects only studied the associations during learning. To control for overall exposure to the pairs, we yoked each of 200 subjects to the exact learning order of subjects in the narrower stimulus set, 48-hr delay experiment. We chose this stimulus set and delay because we were especially interested if the results in the ΔBoth experiment from the otherwise equivalent retrieval-to-criterion experiment would generalize to study-only conditions. Condition affected base pair memory [$F(4,796)$=48.9, $p<0.001$], but here t-tests indicated memory followed a ΔTarget=No Δ=ΔCue>ΔBoth>control pattern (ΔTarget vs. No Δ, p=0.94; ΔTarget vs. ΔCue, p=0.25; No Δ vs. ΔCue, p=0.25; all others, p<0.001) (*Figure 7A*). Therefore, the presence of related associates again aids base pair memory, but in this case, there was very little continued benefit for encountering identical associates while practicing repeated study alone in the No Δ condition (*Karpicke and Roediger, 2008*). In all correlations between retroactive effects and base-secondary pair dependence with relatedness, only ΔTarget retroactive memory correlated with target relatedness (p=0.02) (*Figure 7B*). Correlations in the ΔCue condition were not significant (p=0.85), nor were any clusters along the ΔBoth surface (p=1.0). Intriguingly, correlations between relatedness and memory dependence were not significant in any condition (all p>0.24), suggesting that retrieval during learning may promote more interdependence than study (*Carpenter and Yeung, 2017*).

## Semantic relatedness accelerated new learning

New learning generally benefits from relatedness, whether via associations between words within a pair (e.g., *Nelson et al., 2003*) or with prior learning (*Underwood, 1951*; *Barnes and Underwood, 1959*; *Young, 1955*; *Postman and Parker, 1970*; *Palermo and Jenkins, 1964*; *Jarrett and Scheibe, 1963*; *Wimer, 1964*; *Metcalfe et al., 1993*). Therefore, we also examined the overall effects of

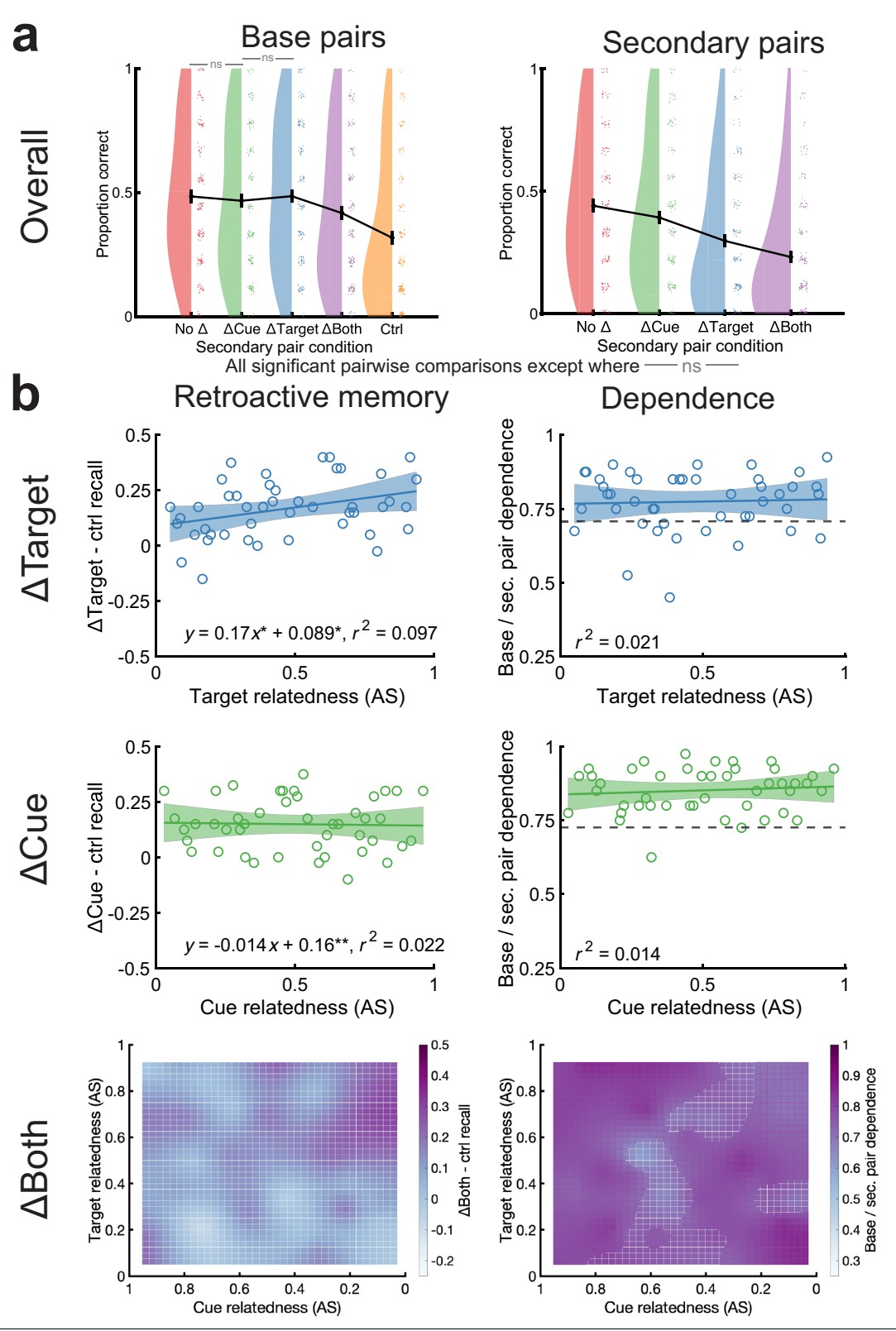

**Figure 7.** Study-only learning led to semantic relatedness benefits in the ΔTarget condition but no base-secondary pair dependence. (**a**) Overall memory performance for base (left) and secondary pairs (right) by condition. All comparisons were significant except those labeled with gray bars and 'ns' (p>0.1) or † (0.05<p<0.1). Data points from individual subjects were jittered slightly for better visualization. (**b**) Correlations between retroactive memory effects (left) and base-secondary pair dependence (right) in the ΔTarget (top), and ΔCue (middle) conditions. Retroactive memory effects

*Figure 7 continued on next page*

*Figure 7 continued*

correlated with target relatedness in the ΔTarget condition, but no other comparisons were significant. Pearson correlations are shown in the plots followed by * when p<0.05 and ** when p<0.01. On bottom, we plotted retroactive (left) and dependence surfaces (right) in the ΔBoth condition.

condition and relatedness levels on secondary pair learning in all experiments employing retrieval-to-criterion learning (the study-only experiment had no learning measure). Accordingly, in the narrower stimulus set experiments, the number of trials to criterion followed a No Δ<ΔCue<ΔTarget<ΔBoth pattern (*Figure 8A*). The wider stimulus set experiments produced a somewhat similar pattern of No Δ<ΔTarget<ΔBoth=ΔCue, where instead ΔTarget and ΔCue flipped from the narrower stimulus set. We next investigated learning across subjects (average trials to criterion per secondary pair) as a function of relatedness in the ΔCue, ΔTarget, and ΔBoth conditions. Higher cue relatedness produced

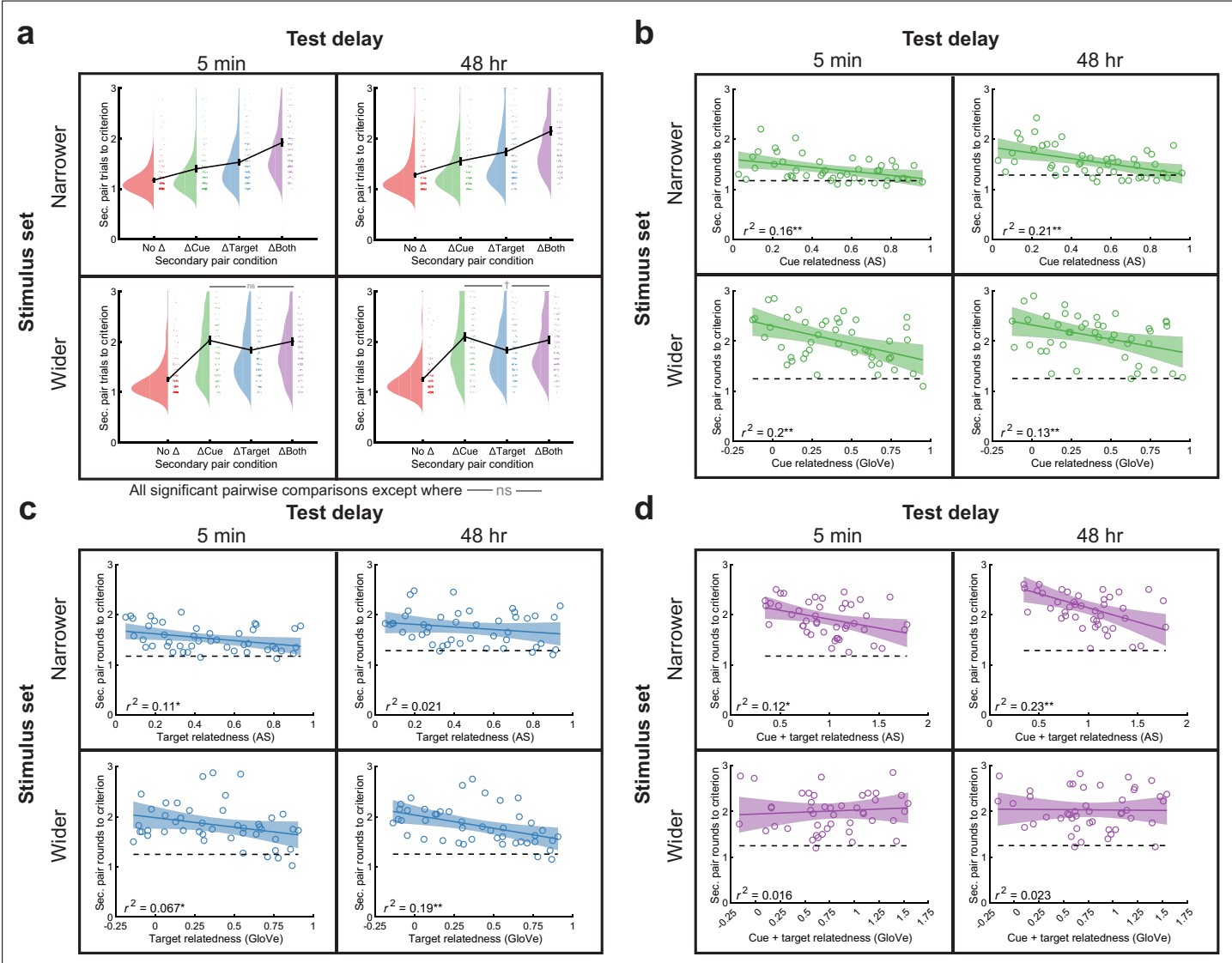

**Figure 8.** Secondary pair learning differed by stimulus set and condition and generally benefitted from semantic relatedness. (**a**) Learning time (mean trials to criterion) followed this pattern for the narrower stimulus set: No Δ 'ns' (p>0.1) or † (0.05<p< 0.1). Data points from individual subjects were jittered slightly for better visualization. (**b**) In the ΔCue condition, average learning time across subjects for each word pair decreased with increasing cue relatedness (top: AS; bottom: GloVe). (**c**) In the ΔTarget condition, learning time generally decreased with increasing B/B' relatedness (top: AS; bottom: GloVe). One exception occurred for the narrower stimulus set, 48-hr delay experiment. (**d**) In the ΔBoth condition, learning time decreased with cue+target relatedness in the narrower stimulus set, but not in the wider stimulus set. In (**b–d**), Pearson correlations are shown in the plots followed by * when p<0.05 and ** when p<0.01.

faster secondary pair learning in every experiment (all p<0.005) (*Figure 8B*). Similarly, higher target relatedness produced faster secondary pair learning in every experiment except the narrower stimulus set, 48-hr experiment (p=0.17; all others p<0.05) (*Figure 8C*). Finally, additive cue+target relatedness generally produced faster secondary pair learning in the narrower stimulus set experiments (*Wimer, 1964*) (5-min delay: p=0.01; 48-hr delay: p<0.001), but not in the wider stimulus set experiments (both p>0.59) (*Figure 8D*). These results suggest again that secondary pairs in the ΔBoth condition in the wider stimulus set are largely treated as new pairs due to the extent of change, as they do not strengthen, nor are they strengthened by, base pairs. Overall, these results suggest that previously learned base pairs scaffold and speed learning of secondary pairs as a function of their relatedness.

Next, we wanted to rule out an alternative possibility raised by these results. Secondary pairs with high relatedness were learned more efficiently, meaning that they had fewer exposures. If the number of exposures increased RI, this would suggest our RF effects could stem in part from lesser interference. We conducted two analyses to address this possibility. First, we correlated new learning efficiency with memorability across pairs in each condition. We found generally weak evidence in favor of this idea, with significant (p<0.05) results in only the ΔTarget condition in the wider stimulus set, 48-hr delay experiment (r=0.30, p=0.02). Second, we ran partial correlations between relatedness and memorability across pairs while controlling for new learning efficiency. These partial correlations remained significant in all of the main analyses above, including in the ΔTarget condition in the narrower stimulus set, 48-hr experiment (r=0.34, p=0.026), wider stimulus set, 5-min delay experiment (r=0.45, p=0.002), and wider stimulus set, 48-hr delay experiment (r=0.36, p=0.016) and for cue+target relatedness in the ΔBoth condition in the narrower stimulus set, 48-hr experiment (r=0.41, p=0.005). Full results from these partial correlations can also be seen in *Supplementary file 9*. Therefore, it appears our RF effects did not rely on the amount of pair exposure during secondary pair learning. We also calculated correlations between base-secondary pair dependence and secondary pair learning efficiency and between dependence and relatedness while controlling for secondary pair learning efficiency. These correlations can be found in *Supplementary file 10*.

## Discussion

We showed that semantic relatedness during learning profoundly benefited memory by retroactively strengthening old associations while scaffolding new ones. We largely found long-term RF across experimental conditions (vs. control), which increased linearly with relatedness in the ΔTarget and ΔBoth conditions. In the stimulus set experiments featuring a wider range of semantic relationships, we found a typical RI effect when relatedness was low and there were short delays after new learning, but both the relatedness of the individual word pairs and the longer delay additively reversed these RI effects into RF. Furthermore, memory dependence (between base and secondary pairs) increased with relatedness in the ΔTarget and ΔBoth conditions. In the paragraphs that follow, we will argue that these results strongly support the theory on recursive reminders, and we next interpret our effects in each learning condition in light of this theory. We follow these interpretations with a possible neurobiological mechanism and a discussion of Osgood's proposal.

Theory on recursive reminders posits that remembering past events during new learning can benefit memory (*Hintzman, 2011*; *Hintzman et al., 1975*; *Otero and Kintsch, 2017*; *Jacoby and Wahlheim, 2013*; *Wahlheim et al., 2019*; *Tullis et al., 2014*; *Benjamin and Tullis, 2010*; *Begg and Green, 1988*; *Smirnov, 1973*). The recursive reminders account has three notable aspects here. First, providing explicit reminders or encouraging subjects to recall or integrate past events just before new learning seems to reduce RI (*Chanales et al., 2019*; *Negley et al., 2018*; *Lustig et al., 2004*; *Huang and Li, 2022*) (and reduce interference in other paradigms; *Anderson and McCulloch, 1999*; *Moeser, 1979*; *Carroll et al., 2007*; *Reder and Anderson, 1980*; *Smith et al., 1978*) or even produce RF (*Wahlheim et al., 2019*; *Burton et al., 2017*). Crucially, subjects must both *notice* a change and *recollect* the changed material for these benefits to arise (*Jacoby et al., 2015*; *Wahlheim and Jacoby, 2012*; *Wahlheim, 2014a*). Second, although these reminders can be directed via instructions, they can also occur spontaneously (*Hintzman, 2011*; *Hintzman et al., 1975*; *Begg and Green, 1988*; *Loftus, 1979*; *Putnam et al., 2017*; *Tousignant et al., 1986*). Relatedly, either strong pre-experimental associations or new episodes linking a target and competitor—which could cause subjects to recollect linked information even when uninstructed—can lower (*Goodmon and Anderson, 2011*) or reverse memory impairment effects (*Chan et al., 2006*; *Bäuml and Hartinger, 2002*; *Anderson et al., 2000*;

*Rowland and DeLosh, 2014*). Third, recursive reminders create interdependence between initial and later-learned information that preserves memory for individual temporal contexts. Interdependence—as measured by whether memories are remembered or forgotten together—can develop for pre-existing or newly learned associations and generally aids memory (*Garlitch and Wahlheim, 2020*; *Ngo et al., 2021*; *Horner and Burgess, 2014*; *Horner et al., 2015*; *Ngo et al., 2019*; *Andermane et al., 2021*; *Wahlheim, 2014b*). Moreover, reminders also scaffold new learning, such that recollecting changes during the formation of new memories aids their retention (*Jacoby et al., 2015*; *Jacoby and Wahlheim, 2013*; *Burton et al., 2017*; *Wahlheim and Jacoby, 2012*; *Wahlheim, 2014a*; *Wahlheim, 2014b*).

Broadly, in accord with the recursive reminders account, we propose that strong pre-existing associations between base and secondary pairs produce retroactive benefits by increasing the likelihood of subjects recollecting base pairs and relating them to secondary pairs, thereby increasing both base pair memory and dependence. Critically, the absence of greater intrusions with higher relatedness suggests that temporal order information regarding the learning list of each pair was preserved rather than the two contexts becoming confused. That is, with high relatedness, recursive reminder-induced memory benefits outweighed the possible countervailing force of competition at retrieval. With weaker pre-existing associations, we speculate that subjects noticed changes while learning new secondary pairs but may have been less likely to recall and integrate them with corresponding base pairs.

We now discuss our results and how they support the recursive reminders account in each condition. The ΔTarget condition showed clear retroactive effects except in the presence of ceiling performance: target relatedness linearly increased RF, including when subjects only studied the information; RI/RF depended on the delay, such that, under the wider stimulus set experiments, RI occurred with a short delay, whereas higher relatedness rescued individual pairs from RI to no effect, and with a longer delay, relatedness increased benefits from no effect to RF; and base pair target-secondary pair target duo dependence increased with target relatedness (except this also did not occur in the study-only experiment). As introduced above, these effects likely stem from two competing processes: a temporary retrieval impairment due to high competition with the more recently learned target word, and a long-term strengthening and interdependence effect that increases linearly with relatedness. High relatedness between base and secondary pairs means that during secondary pair learning, base pairs are reactivated via recursive reminders and secondary pairs become scaffolded to them. These results mirror other phenomena that differentially affect short-term and long-term memory performance, such as the benefits of testing (characterized as the testing effect) (*Roediger and Karpicke, 2006b*; *Bjork and Bjork, 1992*). Although we have largely avoided comparing 5 min and 48 hr delay results directly because they were part of different experiments (and therefore subjects were not randomly assigned to different delays), it appears that conditions in which we expect recursive reminders tend to slow the rate of forgetting (relative to the amount of forgetting in the control condition). This pattern suggests recursive reminders act as effective retrieval processes, similar to findings on the testing effect (*Hintzman, 2011*).

In contrast to the ΔTarget condition, RF tended to occur overall in the ΔCue condition, with null effects in the restudy experiment and the wider stimulus set, 5-min experiment, and featured a weaker relationship with cue relatedness. It is less clear whether and how effectively recursive reminders occur in this condition. In one sense, the absence of a relationship between relatedness and memorability (or the presence of a very weak benefit that remained insignificant in our main analyses) could suggest less recursive reminding and that most benefits occur due to a combination of increased availability of target responses (in the absence of competition at retrieval). Indeed, increasing response availability has been proposed to occur independent of associations (*Martin, 1965*), and rehearsing B responses alone can improve A-B memory (*Estes, 1979*). However, weak relatedness benefits in this condition across all experiments indicate that subjects may perform a mental 'Δcue→cue→target' operation during secondary pair learning that would require recollecting the original association as a recursive reminder. Altogether, the retroactive benefits are clear in the ΔCue condition, but the mechanism seems to differ from the ΔTarget condition and the extent to which recursive reminders are specifically involved versus other processes such as increased target availability is unclear.

Strikingly, in the ΔBoth condition, RF occurred overall (vs. control) and at high values of cue and target relatedness in the narrower stimulus set, 48-hr delay experiment. Moreover, base-secondary

pair dependence similarly occurred at high cue and target relatedness in this experiment. These results suggest that, as in the ΔTarget condition, there was an increased likelihood of recursive reminders supported by pre-existing associations. These effects occurred when cue and target relatedness were high, possibly because changing both causes subjects to fail to recollect both pre-existing links and/or attribute the secondary pair to a new memory (*Shin and DuBrow, 2020*). Additionally, null effects in the wider stimulus set experiments—even for pairs with high cue and target relatedness—suggest that relatedness across pairs may also need to be high on average for subjects to start noticing and recollecting changes rather than attributing the pairs to entirely new associations. We do note that, in the narrower stimulus set, 48-hr delay experiment, cue and target relatedness had somewhat dissociable effects in the ΔBoth condition, such that target relatedness predicted memorability while cue relatedness predicted dependence. We find these results intriguing and worthy of further investigation in future work.

Overall, we propose that relatedness increases the likelihood of recursive reminders, which create well-fortified and interdependent sets of associations that maintain and even strengthen memories while preserving information such as their temporal and contextual order (*Hintzman, 2011*; *Wahlheim and Zacks, 2019*; *Jacoby et al., 2015*). These explanations rely heavily on the importance of interdependencies among base and secondary pairs, a concept elucidated by paradigms featuring multi-element 'closed-loop' learning configurations (*Horner and Burgess, 2014*; *Horner et al., 2015*; *Ngo et al., 2019*) like A-B, B-C, and A-C. These configurations enhance memory and the interdependence between associated elements more than similar, 'open-loop' configurations like A-B, B-C, and C-D. In this framework, our high relatedness ΔTarget and ΔBoth conditions resemble a closed loop, except that rather than having to learn the final link in the loop de novo, unrelated associations are closed by the pre-existing target relationship in the ΔTarget condition and by both pre-existing cue and target relationships in the ΔBoth condition. These results demonstrate a clear interplay between semantic and episodic representations, whereby semantic representations scaffold the formation and retention of episodic memories (*Irish and Piguet, 2013*; *Renoult et al., 2019*). Finally, under ΔTarget learning, testing causes subjects to recollect stimulus changes more often than restudying (*Wahlheim, 2014a*). Therefore, in our study-only experiment, interdependencies may not have emerged because subjects were not forced to incorporate changes occurring between base and secondary pairs into an integrated memory trace, suggesting that testing during learning is critical for forming these interdependencies (*Carpenter and Yeung, 2017*).

Neural results also support the idea that retrieving earlier memories during new learning aids memory and interdependence. Generally, the neocortex supports networks of semantic information, whereas the hippocampus binds together elements specific to episodes (*Horner et al., 2015*; *McClelland, 1995*). Retrieval cues often elicit reactivation of incidental (non-target) information in hippocampus (*Miller, 2013*) or neocortex (*Jonker et al., 2018*; *Horner et al., 2015*) (which in turn coincides with greater hippocampal activity; *Horner et al., 2015*). Reactivation in cortex (*Chanales et al., 2019*; *Koen and Rugg, 2016*) or hippocampus (*Kuhl et al., 2010*) during new learning predicts resistance to interference and inference for information linked by a common element (e.g., A-C after A-B and B-C learning) (*Zeithamova et al., 2012*; *Shohamy and Wagner, 2008*). Moreover, instructions to integrate cause subjects to form neural patterns distinct from ordinary encoding, which predict behavioral measures of integration (*Chanales et al., 2019*; *Richter et al., 2016*). These results all suggest that recollection during new learning reactivates and strengthens old memory traces, promoting resistance to interference and interdependence. Finally, evidence from rodents suggests that neurons encoding prior memories are reactivated upon learning-related experiences (*McKenzie et al., 2013*; *McKenzie et al., 2014*), offering a plausible way in which integration can occur. Furthermore, blocking hippocampal plasticity during new, overlapping events prevents transfer between the two memories, suggesting a *causal* role for the hippocampus in this process (*Iordanova et al., 2011*).

Given these findings, we now speculate on how our effects fit within a neurobiological framework extending the recursive reminders account (*Figure 9*). In our study, pre-existing semantic relationships existed primarily within the neocortex, while previously unrelated pairs were bound by the hippocampus along with their episodic list context and novel, related episodes could also have been interdependently linked within the hippocampus. The No Δ condition generally produced maximum strengthening for both base and secondary pairs. In line with a theory suggesting that even repetitions of the same learning material create multiple traces within the hippocampus (*Nadel and Moscovitch,*

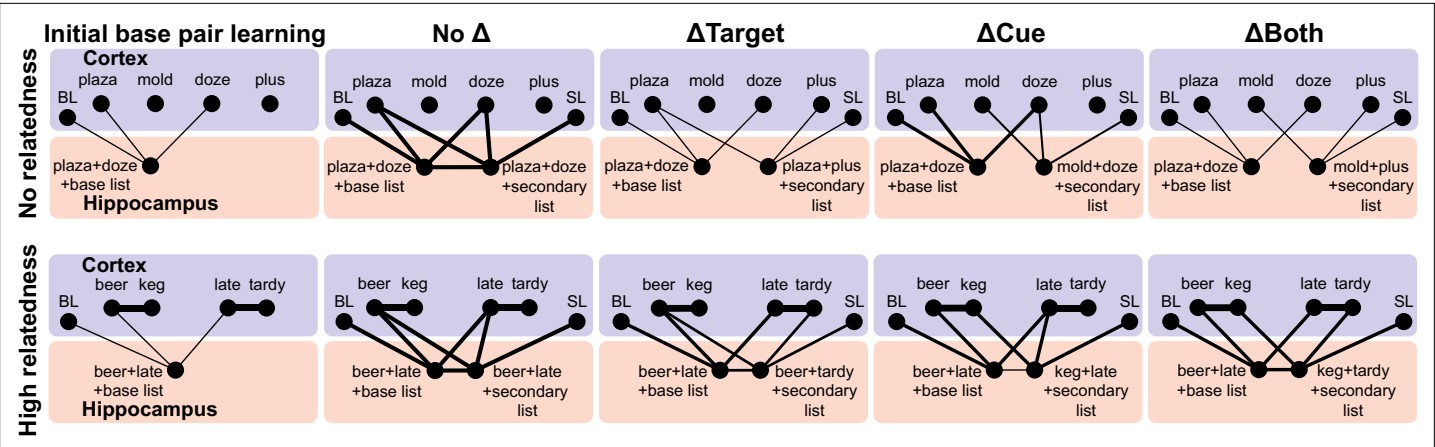

**Figure 9.** Proposed neurobiological schematic of long-term memory fates across conditions and two levels of semantic relatedness. Initially, unrelated base pair associations are bound in the hippocampus along with base list (BL) contexts (e.g., 'plaza-doze-base list' and 'beer-late-base list'; left column), followed by new associations in secondary list (SL) contexts. Subjects can thereby retrieve target words given a cue and list context. Cortical associations are absent for words without pre-experimental relatedness (top) and strong for words with high pre-experimental relatedness (bottom). Relearning A-B (No Δ condition; second from left column) under either level of relatedness results in strengthened base pair and robust secondary pair associations (e.g., 'plaza-doze-secondary list'), likely as linked episodes within the hippocampus. The consequences for learning a pair with a new target (ΔTarget condition; middle column) differs based on pre-experimental relatedness: with no relatedness (top), little to no long-term change occurs to the base pairs and secondary pairs are independently formed (e.g., 'plaza-plus-secondary list'), whereas with high relatedness (bottom), the original memory is strengthened as the new memory is formed (keg-late-secondary list). Moreover, the episodes become bound within the hippocampus, forming an interdependent memory trace. Note that RI effects shown in the narrower stimulus set, 5-min delay experiment are not represented by this long-term schematic and likely stem from more temporary retrieval impairments. Learning a pair with a new cue (ΔCue condition; second column from right) generally strengthens the original memory (perhaps due to better target accessibility) in a manner only weakly reliant on cue relatedness. The consequences for learning a pair with two new words (ΔBoth condition; rightmost column) differ markedly based on pre-experimental relatedness: with no relatedness (top), the new association memory is formed independently, whereas with high relatedness (bottom), the base pair memory becomes strengthened, and the base and secondary pair associations become interdependent. RI, retroactive interference.

1997), these context memories were formed individually yet, as shown by their strong interdependence (red, leftmost point in *Figure 6B*), they were strongly bound within the hippocampus. In the ΔTarget and ΔBoth conditions (and to a lesser extent in the ΔCue condition), high relatedness between base and secondary pairs—represented by pre-existing cortical links—facilitated base pair reactivation via recursive reminders, thus creating an inter-hippocampal association and behavioral dependence. Such benefits were far more limited with no or low relatedness. Additionally, though we propose that these interdependencies initially rely on the hippocampus, the novel associations may eventually become linked within the cortex with more repetitions (*Antony et al., 2017*; *McClelland, 1995*). Altogether, the model in *Figure 9* provides a simplified, speculative explanation for our collected results and a testable mechanism for future studies.

How do our results support or refute the key ideas in Osgood's (*Osgood, 1949*) proposal? The presence of retroactive benefits that increase along the cue identity line strongly support his proposal that RI becomes RF with high target relatedness. However, note that we only found the full crossover from RI to RF across experiments—such as across delays by contrasting both wider stimulus set experiments or across overall levels of relatedness by contrasting the narrower and wider 5-min experiments—rather than finding the crossover within the same experiment. RF (and the total absence of RI) along the target identity line supports his proposal, but conversely, the absent (or very weak) positive relationship with cue relatedness in all experiments does not. Finally, along the bivariate surface, we found benefits when the relatedness among both dimensions was very high, but no reliable effects otherwise; such an effect is present visually on Osgood's proposed surface, but it had lacked empirical support. We also assert that there is no *one* surface, as we produced surfaces that varied both by overall relatedness and delay. Note that we explored a wide range of semantic relatedness in later experiments, but one could consider even further notions of relatedness. When later-learned information differs in even more substantial ways, such as when numbers, letters, or famous people are presented when trying to recall one of the other categories, RI tends to decrease again because the

different categories reduce competition during retrieval (*Bower et al., 1994*; *Smirnov, 1973*; *Skaggs, 1925*). Therefore, if one took the wider stimulus set, 5-min delay surface and expanded stimulus relatedness into domains beyond words, RI may decrease again, producing a nonlinear effect reminiscent of those found across early studies employing a wide variety of stimulus types, including with numbers, number-letter combinations, nonsense syllables, abstract visual symbols, poetry, and prose (*Britt, 1935*; *Robinson, 1927*; *Gibson, 1941*; *Parducci and Knopf, 1958*; *Cheng, 1929*; *Lund, 1926*; *Dreis, 1933*; *Harden, 2010*; *Gibson and Gibson, 1934*; *Rothkopf, 1957*). Additionally, future studies could explore how surfaces differ based on less nameable stimuli types, such as stimulus spaces with continuous, quantifiable changes in visual stimuli (e.g., *Wammes et al., 2021*; *Molitor et al., 2021*; *Iordan et al., 2020*; *Natu et al., 2016*), or other domains (*Dennis, 1976*).

A notable limitation of our study is that we did not dissociate between semantic representations based on distributed representational models (e.g., *Pennington et al., 2014*; *Plaut, 1995*) and associative relations based on association strength and spreading activation (e.g., *Nelson et al., 1998b*), instead relying on the broad term, semantic relatedness, to capture both constructs. Analyses in *Supplementary file 6* did not indicate any clear distinction in correlations with memorability and interdependence between measures of semantic and contextual similarity such as LSA and word2vec versus AS. Nevertheless, we did not aim to directly contrast these two constructs by selecting stimuli that dissociate them, and given that the two types of relations show different effects on cognition (e.g., *Thompson-Schill et al., 1998*), this will be important in future work. Another limitation to our interpreting these results within a recursive reminders framework is that we did not directly manipulate instructions (e.g., *Jacoby et al., 2015*) or collect in-the-moment responses regarding whether subjects engaged in recursive reminders (e.g., *Wahlheim et al., 2019*). Such manipulations constitute important future directions.

We began this investigation by asking two fundamental questions about learning: when does a new memory facilitate versus interfere with an older one, and when do their fates become linked? Inspired by a never-fully-tested, seven-decade-old proposal (*Osgood, 1949*), we produced a consolidated account suggesting that semantic relatedness between old and new memories facilitates old memory strength and promotes their interdependence. When a new memory bears high relatedness to an old one, subjects can readily notice the change and recollect the old memory, fortifying the old memory and scaffolding the new one to it, providing mutual reinforcement (*Floyd and Goldberg, 2021*). To return to our opening example, if after learning about sweet vermouth, the bartender notices and recollects the change when learning about dry vermouth, the sweet vermouth memory becomes strengthened, and the vermouth memories become linked. Ultimately, these findings highlight and strongly reinforce the importance of building on prior knowledge in educational (*van Kesteren et al., 2012*) and aging (*Badham et al., 2012*) domains and clarify when and how complex networks of knowledge can be accumulated and retained.

## Materials and methods
### Subjects

For each experiment, we chose a sample size of 200—large relative to most memory studies—to ensure that memorability measures from each condition would have reliable data given the fivefold counter-balance (200 subjects/5 counterbalances=40 measurements for each independent word pair contrast). All subjects across the five experiments were undergraduate students with normal or corrected-to-normal vision who received psychology course credit for participating. In each experiment, subjects with overall memory performance less than 4 standard deviations below the mean were excluded and subjects were run until 200 remained. In the study-only experiment, we additionally dropped subjects with no correct responses in the base pair condition, even though it was within 4 standard deviations of the mean. Additionally, numerous subjects did not return or complete the final test. The final break-downs were as follows: narrower stimulus set, 5-min delay: N=201, 0 no test, 136 females (135 post-exclusions); narrower stimulus set, 48-hr delay: N=208, 6 no test, 114 females (108 post-exclusions); wider stimulus set, 5-min delay: N=206, 0 no test, 101 females (99 post-exclusions); wider stimulus set, 48-hr delay: N=212, 7 no test, 149 females (139 post-exclusions); study-only: N=226, 6 no test, 177 females (156 post-exclusions). In the original (retrieval-to-criterion) narrower stimulus set, 48-hr experiment, subjects took the experiment on lab computers. Later experiments were conducted

online due to the COVID-19 pandemic. Timing in the online experiments was identical, and subjects took the experiment while in virtual sessions with research assistants to enhance attentiveness to the task. All subjects were recruited via an online scheduling software. Subjects provided informed consent, and all procedures were in accordance with the California Polytechnic State University, San Luis Obispo Institutional Review Board.

### Stimuli

Since there are numerous word attributes (e.g., word frequency) by which verbal memorability differs (*Rubin and Friendly, 1986*; *Madan, 2019*; *Greene and Tussing, 2001*; *DeLosh and McDaniel, 1996*; *Xie et al., 2020*), we designed our experiments so that the main measure of interest (base pair memory) used the same 45 word pairs, with the only manipulations being the condition of secondary pair learning and the semantic relatedness between base and secondary pairs.

Stimuli selection proceeded in two stages: first, we found paired words of varying relatedness that would later become a cue-Δcue unit or target-Δtarget unit, and second, we created pairs of pairs (comprising a cue, Δcue, target, and Δtarget word). In the first step in the stimulus set with a narrower range of semantic relationships, we aligned 3–5 letter words by free AS from an open database (*Nelson et al., 1998a*) and selected 90 unique pairs ranging approximately evenly from 0.03 (pious→holy) to 0.96 (moo→cow). Note that AS relationships are directed and can affect cued recall memory (*Caplan et al., 2014*; *Popov et al., 2019*), so we controlled for these asymmetries by always designing the cue word from the database (e.g., moo) to the secondary pair (Δcue or Δtarget) and the target word (e.g., cow) to the base pair (cue or target). Our logic was that secondary pair learning would thereby retroactively 'act upon' base pair learning in a predictable way (based on the AS value) rather than vice versa. In the first step in the stimulus set with a full range of semantic relationships, we derived pairwise Global Vector (GloVe) cosine similarity [$\cos(\theta)$] values (*Pennington et al., 2014*) for all 3–5 letter words from the same free association database. The GloVe training set involved 840 billion web tokens and was imported using the gensim Python toolbox (https://github.com/RaRe-Technologies/gensim) (*Rehurek and Sojka, 2010*). We then chose words to quasi-evenly span the full interval of $\cos(\theta)$ values ranging from –0.14 to 0.95. These relationships are undirected and therefore the base versus secondary pair decisions were somewhat arbitrary but remained consistent for all subjects after initial determination.

Within each experiment, we aimed for each subject to experience similar overall levels of semantic relatedness across pairs. Therefore, in the second step of creating the stimuli, we separated all 90 chosen pairs into thirds by semantic relatedness (30 pairs each) and randomly assigned 30 pairs from each split into 15 cue pairs (cue and Δcue) and 15 target pairs (target and Δtarget). Then we randomly assigned the 15 cue pairs from each relatedness level to one of three target pair levels and vice versa, meaning that all pairs were essentially assigned to ninths of a 2-D grid (with 5 cue and 5 target pairs in each ninth). Effectively, these ninths could be classified by low-moderate-high (l/m/h) semantic similarity for cues and targets, respectively, as l/l, l/m, l/h, m/l, m/m, m/h, h/l, h/m, and h/h. Next, we randomly paired the pairs within each ninth so that each had five pairs of pairs consisting of cue, Δcue, target, and Δtarget words that could be assigned to any experimental condition. Our counterbalance separated one item from each ninth into each of the five experimental conditions and therefore multiples of five subjects were required to maintain counterbalances across stimuli.

### Other relatedness metrics

Forward (cue→Δcue and target→Δtarget) AS and backward mediator strength values (e.g., cue←[other word]←Δcue) were taken from the same repository (*Nelson et al., 1998a*) from which we found cue←Δcue and target←Δtarget AS values used in the main analyses. Weighted path length was determined by first finding the shortest path in a network composed of all words within the repository from target to cue word and then adding up the weights. For example, if the Δcue word was 'stripe' and cue word, 'king,' the shortest path may be 'stripe→tiger (0.034 AS)→lion (0.308 AS)→king (0.021).' Each weight was computed as 1–AS, so the previous example would have a weighted path length of [(1–0.034)+(1–0.308)+(1–0.021)]=2.637. Note that with all single-step associations in the narrower stimulus set, weighted path lengths were all simply 1–AS. Despite the vast size of semantic networks, they possess small-world architecture with generally small path lengths (*Steyvers and Tenenbaum, 2005*), and indeed all but six pairs had lengths of less than 6.

These remaining path lengths had infinite length according to our algorithm and were set to 6. For the spreading activation analysis, we performed the following, starting with the target node: (1) at each node, find all edges (to nearby words in free association space) and norm all AS to 1, (2) find activation (if any) of the cue word and log it by its weight, (3) advance along all edges iteratively, and (4) repeat up to three steps, logging as weights the multiplied values of each edge en route to the cue word. Then we added up the total activation of the cue across these three steps. For word2vec (*Mikolov et al., 2013*), we used a version of the model trained on Google News with 3 million 300-dimension word vectors. We imported the vectors and calculated similarity using the gensim Python toolbox (*Rehurek and Sojka, 2010*). For latent semantic analysis, we found pairwise cosine similarity values via http://lsa.colorado.edu using term-to-term comparisons trained on general reading lists up to the college level with 300 orthogonal factors (*Landauer and Dumais, 1997*).

## Procedure

All experiments followed this order: base pair learning, secondary pair learning, a 5-min or 48-hr delay, base pair testing, and secondary pair testing (*Figure 1A*). For base pair learning, subjects first viewed the 45 pairs in a round of encoding followed by retrieval to criterion or, in the case of the study-only experiment, repeated study. During encoding, subjects attended to a fixation cross for 1 s before pairs appeared for 4 s. Cue and target words were shown just above and below the vertical center of the screen, respectively, and both were centered horizontally. During retrieval in the main retrieval-to-criterion experiments, subjects attended to a fixation cross for 1 s before the cue word was shown. After 1 s, a blank prompt was shown where subjects could type in their answer. Subjects were given unlimited time to respond, after which both cue and target words were shown as feedback whether correct or not. Correct pairs were dropped from this phase so that the only remaining trials on successive rounds were previously incorrect responses. In the study-only experiment, subjects were given 4 s to restudy the words rather than testing.

Before secondary pair learning, we told subjects they would next learn a new list of pairs and that the stimuli may or may not change between lists. Secondary pair learning then proceeded with the same retrieval criterion and timing as base pair learning, except that only 36 pairs were learned (corresponding to pairs in the No Δ, ΔTarget, ΔCue, and ΔBoth conditions).

In the study-only experiment, the yoking procedure for each subject was matched to the learning order of a subject from the retrieval-to-criterion, narrower stimulus set, 48-hr delay experiment. For example, subject #1 from the retrieval-to-criterion experiment was matched with subject #1 from the study-only experiment. We copied the learning order precisely in both base and secondary pair learning phases from each retrieval-to-criterion subject to the study-only subject, so if subject #1 in the former saw 'sick-push' first, so did subject #1 in the latter experiment, and so on. In later rounds of learning, when many word pairs had dropped out for a subject in the retrieval-to-criterion experiment, those same items dropped out for the yoked subject in the study-only experiment. We acknowledge that this procedure cannot control for individual differences in memorability; for instance, subject #1 in the retrieval-to-criterion experiment and subject #1 in the study-only experiment could require different exposures to achieve the same criterion, so the pairs remaining at the end of each learning phase may not be the ones the study-only subject would have struggled to learn. Such differences cannot be addressed without an additional test assessment in the study-only experiment. We believe that matching the objective amount of exposure was the best way to match restudy to retrieval-to-criterion learning conditions, but another experiment using a fixed number of repetitions for each pair in restudy and test conditions or an experiment that randomly assigns the number of repetitions across pairs in restudy and test conditions would address the role of individual differences more precisely.

After returning for the test, subjects were first asked to recall all words from the base list (which we described as the first list they learned). During the test, subjects attended to a fixation cross for 1 s before the cue word was shown. They were allowed unlimited typing time and were given no feedback after submitting their response. Following one test of each pair, they were then asked to recall all words from the secondary list (which we described as the second list they learned), which followed an identical format. Subjects were then debriefed and allowed to leave.

## Statistics

Across-condition comparisons within each experiment were conducted using one-way (condition: No Δ, ΔTarget, ΔCue, ΔBoth, and control), repeated-measures ANOVAs. For comparisons in which the sphericity was violated, we corrected the degrees of freedom in the F-ratio test using Huynh-Feldt correction. Significant ANOVAs were followed with pairwise, FDR-corrected (*Benjamini and Hochberg, 1995*), within-subject t-tests.

Across-base pair memorability analyses were conducted first by finding the proportion of subjects who recalled each base pair in each condition. For instance, if subjects #2, 7, 12, and 17 had 'copy-angel' in the ΔTarget condition and 3/4 recalled it, while subjects #5, 10, 15, and 20 had 'copy-angel' in the control condition and 2/4 recalled it, the ΔTarget – control memorability for that pair would be 0.25. Next, we ran ordinary least squares (OLS) linear regression analyses between the memorability of that pair and its specific AS or GloVe cos($\theta$) value. For the ΔBoth condition, we added cue+target relatedness values before conducting the regression. In some cases, the y-intercept of the OLS line may be theoretically meaningful (e.g., 0 AS means 0 subjects might endorse a word in a free association task). In all cases, the slope was of interest, indicating whether and how relatedness affected condition-based memorability. Significance for both slope and intercept results were reported in each plot based on p values from 'fitlm' in MATLAB, and best-fit lines were plotted with the confidence error output from 'polypredci' in MATLAB (*Strider, 2021*).

Memory dependence was calculated by first examining memory for each base pair target-secondary pair target duo within a condition across subjects. Consider the following example in the ΔTarget condition, where 1=correct and 0=incorrect memory and values are represented respectively across subjects. If answers for subjects #2, 7, 12, and 17 when given 'peace' as the cue during the base pair test (correct answer: 'razor') were 1, 1, 1, and 0 and their answers when given 'peace' as the cue during the secondary pair test (correct answer: 'shave') were 1, 1, 0, and 0, the across-subject dependence would be 0.75. Conversely, if the base pair test across the same subjects for the same pair was 1, 1, 1, and 0 and the secondary test was 0, 1, 0, and 1, the across-subject correlation would be 0.25. It is important to note that, between the extremes of ceiling and floor performance, dependence and performance are dissociable. A target duo could potentially have a higher rate of dependence than raw memory performance if it regularly became forgotten together. Conversely, a target duo could have a lower rate of dependence than memory performance if one of the two pairs are regularly recalled and the other not. Following this calculation, we performed similar regression analyses with these values plotted against semantic relatedness in each experiment. Additionally, we determined thresholds for each word pair by finding the dependence of each cue-target pair versus all other mismatched, cue-Δtarget pairs. Since there were 44 other pairs, our upper threshold was whether the true pair was higher than 43/44=0.977 of the other pairs, which corresponds to a significance threshold of α=0.046. We plotted the average of these thresholds within each experiment as a dotted line on each dependence graph.

To create the memorability surfaces in the ΔBoth condition, we first examined the ΔBoth – control condition memorability across subjects for each base pair (as above). Next, we found the bivariate cue and target relatedness for each base pair. From this, each pair had three coordinates: a cue relatedness value (which would become the y-coordinate on the surface), a target relatedness value (the x-coordinate), and a ΔBoth – control memorability value (the z-coordinate). To obtain a smoothed surface from these data, we used robust spline smoothing of the z values over the x-y surface with a smoothing factor of approximately 40% of the input space (e.g., 0.37 for data spanning 0.93 of AS values and 0.43 for 1.09 of cosine similarity values) (using the 'smoothn' function in MATLAB) (*Garcia, 2010*). This smoothing factor was used to cover inevitable gaps in the surface space (see *Figure 1D* for illustration of this point). Above- and below-zero thresholds in the surface space were calculated like the above using bootstrapped permutation tests, and we similarly smoothed over these surfaces using the same smoothing factor. To assess significance, we found the sizes of 2-D clusters where the true values exceeded the above-zero thresholds. Because a noisy signal could exceed this threshold simply by chance, we next computed the likelihood of finding a cluster of the observed size. To do this, we used 1000 permutation tests, whereby we randomly scrambled whether an across-subject memorability value fell in the ΔBoth or control condition, and we found the size of each of these clusters exceeding the above-zero permutation threshold. Finally, we obtained a p value by examining the proportion of permutation tests that our observed cluster exceeded.

## Acknowledgements

The authors would like to thank Anna Leshinskaya and Charan Ranganath for helpful discussions about the project and Xiaonan Liu and Sebastian Michelmann for comments on early drafts of the manuscript. The authors would also like to thank several R.A.s who helped to run the study, including Lauren Hansen, Lily Sanz, Jacob Van Dam, Kenia Alba, Annika Asp, Kaeley Benedict, Kirrin Bereznak, Nicole Brault, Kylie Capella, Rasha Demeter, Noa Dunevich, Chloe Fleischer, Lauren Garabedian, Samantha Garrett, Shana Gitterman, Olivia Gott, Trevor Guerra, Mackenzie Harrison, Ethan Heh, Erika Holloway, Caitlin Johansen, Jarett Massey, Katherine Miller, Rachel Nebel, Sahar Oliaei, Catherine Palmer, Madeline Phillips, Natalie Phillips, Matthew Reed, Pilar Reyes, Sofía Sanz Galan, Isabella Strawn, Arushi Tewari, Natalie Thomas, Alyssa Tierney, Sarah Tung, and Emma Whitwam. This work was supported by the Princeton University CV Starr Fellowship to JWA.

## Additional information

### Funding

| Funder | Grant reference number | Author |
|---|---|---|
| Princeton University CV Starr Fellowship | | James W Antony |

The funder had no role in study design, data collection and interpretation, or the decision to submit the work for publication.

### Author contributions

James W Antony, Conceptualization, Data curation, Formal analysis, Methodology, Software, Supervision, Validation, Visualization, Writing – original draft, Writing – review and editing; America Romero, Anthony H Vierra, Rebecca S Luenser, Investigation; Robert D Hawkins, Formal analysis, Methodology, Writing – review and editing; Kelly A Bennion, Investigation, Project administration, Supervision, Writing – review and editing

### Author ORCIDs

James W Antony (iD) http://orcid.org/0000-0003-0656-2170

### Ethics

Informed consent was obtained from each subject based on a form authorized by the California Polytechnic State University, San Luis Obispo Institutional Review Board: #2020-068-CP, "Memory for Emotional and/or Neutral Information".

### Decision letter and Author response

Decision letter https://doi.org/10.7554/eLife.72519.sa1
Author response https://doi.org/10.7554/eLife.72519.sa2

## Additional files

### Supplementary files

• Supplementary file 1. Cue-Δcue and target-Δtarget pairs and their relatedness factors for the narrower stimulus set. AS: associative strength; MS: mediator strength; Spath: weighted shortest path; SpAct: spreading activation strength; GloVe: global vector $\cos(\theta)$; w2v: word2vec $\cos(\theta)$; LSA: latent semantic analysis $\cos(\theta)$. Bold: relatedness factor of interest. Related to *Figure 1*.

• Supplementary file 2. Cue-Δcue and target-Δtarget pairs and their relatedness factors for wider stimulus set. AS: associative strength; MS: mediator strength; Spath: weighted shortest path; SpAct: spreading activation strength; GloVe: global vector $\cos(\theta)$; w2v: word2vec $\cos(\theta)$; LSA: latent semantic analysis $\cos(\theta)$. Bold: relatedness factor of interest. Note that AS and MS values of 0 reflect the absence of a one- or two-step semantic association, respectively. Related to *Figure 1*.

• Supplementary file 3. Overall base and secondary pair memory by condition. Results shown for every experiment and condition with mean and SEM for base (top) and secondary pair (bottom)

memory. Related to *Figure 2* and *Figure 2—figure supplement 1*.

• Supplementary file 4. Correlations among the relatedness factors for the narrower stimulus set after concatenating cue-Δcue and target-Δtarget lists together. AS: associative strength; MS: mediator strength; Spath: weighted shortest path; SpAct: spreading activation strength; GloVe: global vector cos($\theta$); w2v: word2vec cos($\theta$); LSA: latent semantic analysis cos($\theta$). Bold: significant relationships ($P<0.05$). These values are specific to this data set and will vary across data sets or for a full dictionary. Note that the perfect correlation between AS and Spath reflects the fact that Spath is simply (1 – ←AS) with all one-step associations. Related to Results section.

• Supplementary file 5. Correlations among the relatedness factors for the wider stimulus set after concatenating cue-Δcue and target-Δtarget lists together. AS: associative strength; MS: mediator strength; Spath: weighted shortest path; SpAct: spreading activation strength; GloVe: global vector cos($\theta$); w2v: word2vec cos($\theta$); LSA: latent semantic analysis cos($\theta$). Bold: significant relationships ($P<0.05$). These values are specific to this data set and will vary across data sets or for a full dictionary. Related to Results section.

• Supplementary file 6. Correlations between various relatedness factors and base pair memorability (top) and base-secondary pair memory dependence (bottom). For the ΔBoth condition, we used the added cue +target relatedness featured in *Figure 5—figure supplement 1*. Relationships within each column align with the corresponding measures for that condition (e.g., ← AS indicates target←Δtarget AS in the ΔTarget condition). AS: associative strength; MS: mediator strength; Spath: weighted shortest path; SpAct: spreading activation strength; GloVe: global vector cos($\theta$); w2v: word2vec cos($\theta$); LSA: latent semantic analysis cos($\theta$). Bold: significant relationships ($P<0.05$). Related to Results section.

• Supplementary file 7. Correlations between base pair memorability and base-secondary pair dependence for all experiments and conditions. Bold: significant relationships ($P<0.05$). Related to Results section.

• Supplementary file 8. Correlations between various relatedness factors and base pair memorability while controlling for base-secondary pair dependence (top) and various relatedness factors and base-secondary pair memory dependence controlling for base pair memorability (bottom). For the ΔBoth condition, we used the added cue +target relatedness featured in *Figure 5—figure supplement 1*. Relationships within each column align with the corresponding measures for that condition (e.g., ← AS indicates target←Δtarget AS in the ΔTarget condition). AS: associative strength; MS: mediator strength; Spath: weighted shortest path; SpAct: spreading activation strength; GloVe: global vector cos($\theta$); w2v: word2vec cos($\theta$); LSA: latent semantic analysis cos($\theta$). Bold: significant relationships ($P<0.05$). Related to Results section.

• Supplementary file 9. Correlations between base pair memorability and secondary pair learning efficiency (top) and partial correlations between base pair memorability and semantic relatedness controlling for secondary pair efficiency (bottom). For partial correlations, we used backward AS for the narrower stimulus set and GloVe for the wider stimulus set. Bold: significant relationships ($P<0.05$). Related to Results section.

• Supplementary file 10. Correlations between base-secondary pair dependence and secondary pair learning efficiency (top) and partial correlations between base-secondary pair dependence and semantic relatedness controlling for secondary pair efficiency (bottom). For partial correlations, we used backward AS for the narrower stimulus set and GloVe for the wider stimulus set. Bold: significant relationships ($P<0.05$). Related to Results section.

• Transparent reporting form

## Data availability

All code and data are available at https://osf.io/hmj8b/.

The following dataset was generated:

| Author(s) | Year | Dataset title | Dataset URL | Database and Identifier |
|---|---|---|---|---|
| James A | 2022 | Semantic relatedness retroactively boosts memory and promotes memory interdependence across episodes | https://osf.io/hmj8b/ | Open Science Framework, hmj8b |

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
