## [Editor Report]

The study addresses a classical question of the complex dynamics of long term (semantic) memory and episodic learning, using a impressive behavioral data set, revealing the specific interactive patterns between old and new memories. It should have broad implications to how we study learning and memory in general.

---

## [Decision Letter]

**Decision letter after peer review:**

Thank you for submitting your article "Semantic relatedness retroactively benefits memory and promotes memory interdependence across episodes" for consideration by *eLife*. Your article has been reviewed by 3 peer reviewers, one of whom is a member of our Board of Reviewing Editors, and the evaluation has been overseen by a Reviewing Editor and Chris Baker as the Senior Editor. The following individual involved in review of your submission has agreed to reveal their identity: Brice Kuhl (Reviewer #3).

Essential revisions:

1) Provide a clear, coherent theoretical framework to clarify the design motivation, results, interpretation, and conceptual advances being made. While all reviewers were highly positive about the empirical rigor, they all had reservations about the theoretical framing. R1 and R2 commented on the confusion of the multiple theoretical accounts throughout and that it is not clear how your results support or refute or distinguish among theoretical alternatives; R2 raised specific, excellent point about the problem of recursive reminder to explain some data points; R3 also commented the lack of significant conceptual advances and the limited influence outside human episodic memory. The paper needs to be substantially revised to improve on the theoretical framework to address these major concerns.

2) The motivation of the specific, central, manipulation choices (narrow vs. wide; measuring semantic relatedness using "strength of association") are currently unclear. Addressing this point may rely on having a clear theoretical framework and may also help address the concern of limited influence raised in Point 1 above. That is, at least those who are interested in the dynamics of semantic memory and the interface between semantic and episodic memory (or learning in general) may also find your results relevant once the semantic manipulations are clearly explained.

3) Some methodological details need to be addressed by additional analyses and/or clarification: the confound of the repetition time differences, how yoking subjects to the same learning orders were done, and how dependence scores were computed.

*Reviewer #1 (Recommendations for the authors):*

The study tackles the memory dynamics by behaviorally examining what happens to the old (associative) memory when new (associative) memories are formed, and how it is affected by semantic relatedness (prior, long term, semantic memory). Through a grand experiment with 1000 participants and systematic manipulations of semantic relatedness across cue and target, broader semantic context, and delay between old and new associative learning, they provided a rich and robust empirical dataset showing how semantic relatedness between old and new learning in general strengthen the old memories and the old-new relatedness. A cognitive framework memory dynamic is presented to account for the data.

Clearly depicting the behavioral profiles of the dynamic relationships among long term (semantic/associative) memory and the new memory is much desired for the understanding of the complex memory formation and storage. The careful experimental manipulation, the impressive sample size (both subject and stimuli) and the data analyses and visualization, are laudable. The consideration of multiple types of semantic distance measures (internal vs. external) also helps establish the robustness of the effect. The value of the empirical results aside, I have concerns about several aspects of the current manuscript.

The paper is hard to read, not only because of the richness of the conditions, but mostly because of the lack of a clear and coherent theoretical framework to guide the design and interpretation. The Introduction states that it tests the classical Osgood proposal and predictions were made on that model. But no clear alternatives are analyzed, and it is difficult to appreciate the implications of the results consistent or inconsistent with the proposal. The discussion presented several additional accounts/explanations (e.g., recursive reminders, neural models), but they do not seem to constituent explanations but rather descriptions of the data. What is the relationship between these proposals? What specific assumptions about the memory system is supported/refuted by the data? More specifically, while the different effects of cue- and target- relatedness, semantic context (wide/narrow), and temporal delay are certainly intriguing, they are not clearly motivated or explained. Thus, how exactly memory models are constrained by these empirical findings are not easily appreciated.

A further concern regards the treatment of "semantic effects/semantic relations". The study distinguishes old and new memory, the "semantic effects" they observed are part of an "older" memory. The way the authors use "semantic relatedness" to describe the effect of "prior knowledge" does not make distinctions between potentially different types of long-term relations in the memory representations such as semantic vs. associative -- in the main analyses the semantic distance were operationalized using association strength. It's okay to be parsimonious and show that the distinction is not necessary, and they affect memory dynamics in the same way. But this needs to be explicitly discussed, given the psycholinguistic literature in the past decades that these two kinds of relations show different effects in comprehension (priming) and production (e.g., picture-word interference) tasks. More generally, the mechanisms underlying associative and semantic memory/learning, and between episodic and semantic memory are expected to be discussed in greater depth and clarity (e.g., see Renoult et a., 2019 TICS). It is also not clear what the psychological reality of one step or multiple steps of semantic relations.

In summary, I appreciate the empirical rigor of the study and find the results potentially very important. But they need to be situated in clearer theoretical framework to show their implications for how memory works and to guide further cognitive/neural studies of memory.

L69 ". In studies in which new and old cues and targets were 70 semantically related, or new cues were related to old cues while new targets were not related to 71 old targets, RI for the original A-B association has been found on the order of minutes (31-34). 72 However, to our knowledge, no study has investigated long-term effects when both new and old 73 cues and targets are related, or with very high levels of semantic relatedness."

- I find this sentence very hard to follow.

*Reviewer #2 (Recommendations for the authors):*

This study has several strengths. It did a comprehensive, vigorous hypothesis test, by examining multiple experimental conditions under the same paradigm and with a large sample size. These results could help to reconcile the mixed findings in the literature. Their treatment of semantic relatedness is also thorough and appropriate. The conclusion is mostly accurate and justified. There are some methodological details that I think should be considered. Finally, the lack of neural evidence somewhat limits its conceptual advances.

My main suggestions are related to the methodological details, which I will detail below.

First, the current study used a retrieval-to-criterion learning paradigm, which is very efficient in achieving the desired learning outcome. Although I think this is appropriate for the initial learning of base pairs, it might introduce additional variances during the learning of second pairs that may complicate the explanations. As depicted in Figure S7, semantic relatedness facelifted new learning, resulting in fewer repetitions for strongly related pairs than weakly related pairs. Since this new learning also influences old memory, it is unclear whether the different repetitions also matter, in addition to semantic relatedness itself.

Second, the comparison between restudy and retrieval practice is very interesting and informative. The authors also made a good effort to match the exposure, by yoking each of 200 subjects to the exact learning order of subjects in the narrower stimulus set, 48-hr delay experiment. However, it is not clear how this was achieved, given the individual differences in the overall memory performance. This could be more complicated considering the memorability of a given pair also differs across subjects.

Third, the examination of memory dependence is also very interesting. I am curious how the dependence score is dissociated with the memory score? Still, one can imagine that this index might be less meaningful for subjects who performed very badly or very well in both tests, as their high dependence score might not reflect the true memory integration.

Fourth, the motivation of some experimental manipulations needs more justifications. For example, why did the author manipulate a narrower vs. a wider range of semantic relatedness? For the restudy vs. retrieval comparison, why the narrower stimulus set, 48-hr delay condition was selected?

My final comment is related to the theoretic account of the results. It seems that the interpretation of the results primarily relies on the recursive reminders account. Although I agree this is a reasonable choice, there are some issues that are worth mentioning. For example, this account might not explain the results under the ΔCue condition very well, and it is unclear whether these effects were mainly driven by the ΔTarget under the ΔBoth condition. It is also not readily transparent to me how it could account for the effect of study-test interval on the overall RI vs. RF patterns. Still, the authors did not discuss how it could account for the restudy vs. retrieval differences. Moreover, since there is no neural evidence to actually examine the reactivation during learning, it is my opinion that the current study did not provide direct evidence to support or refute this account, which somehow limits the conceptual advances of the current study. For the same reason, I am not sure if the neural model proposed in the discussion (Figure 7) is justified by the data.

Although this study used a large sample, some of the effects seem to be unreliable. For example, in figure S8, when both the narrow and wide conditions were merged, the effect of semantic relatedness show different trends, i.e., the regression line did not overlap. Did the authors also examine the strength of semantic association between the two words in each pair, and whether this could be a confounding factor that should be controlled in the analyses?

*Reviewer #3 (Recommendations for the authors):*

Overall, this is a very solid and rigorous paper. The main contribution is to understanding the factors that determine behavioral measures of retroactive interference versus facilitation. In particular, when cue-target associations change (e.g., A-B to A-D), the question is how the degree of similarity between original and new cues and original and old targets will influence the relative interference vs. facilitation effects. This issue has a very long history in experimental psychology. The paper takes particular inspiration from a very influential idea from Osgood that interference/facilitation effects can be described along a continuous surface where cue similarity and target similarity are independent dimensions. The paper generally supports the ideas proposed by Osgood, but represents what is likely the most thorough empirical test of these ideas. Notably, whereas Osgood's original ideas were inspired by combining findings from many different experiments/papers/researchers, the current study packs all of the key experimental variables into a single, cohesive set of experiments.

Other strengths of the paper include relatively large sample sizes, consideration of the effects of delay (5 min vs. 48 hours) and consideration of the effects of retrieval practice vs. study. Additionally, although not a major point of emphasis in the paper, one of the interesting findings is that similarity between original and new items can protect older memories without increasing confusability (intrusions). This perhaps represents the most surprising result in the paper as the alternative prediction would be very reasonable (that semantic relatedness would increase intrusions). It seems difficult to pin down exactly why relatedness was protective against intrusions, but it is an intriguing result.

The biggest limitation in the paper is the amount of conceptual advance. As the authors note, Osgood's original ideas anticipated (most of) the key results in the current study-namely, the idea that facilitation and interference are a function of the similarity between original and new information. While the current paper provides a much more definitive investigation of this than is currently available in the literature, there have also been many papers since Osgood (nicely reviewed by the authors) that anticipate aspects of the current results. In particular, studies looking at integration or recursive reminders have clearly established that processes following initial encoding (of the base pairs) will impact the degree of interference and can even promote facilitation. In the current study, there is no direct evidence that integration or reminders occurred, but it is inferred that relatedness triggered reminders.

Taken together, the paper is likely to be of interest to those that study episodic memory in humans (and, in particular, those interested in memory interference). Given the influence of Osgood's original paper in the field of human learning and memory, this paper may also have substantial influence as a definitive test of Osgood's ideas. However, it is less clear whether the paper will have substantial influence outside this field.

[Editors' note: further revisions were suggested prior to acceptance, as described below.]

Thank you for resubmitting your work entitled "Semantic relatedness retroactively benefits memory and promotes memory interdependence across episodes" for further consideration by *eLife*. Your revised article has been evaluated by Chris Baker (Senior Editor) and a Reviewing Editor.

The manuscript has been improved but there are some remaining issues that need to be considered, as Reviewer 2 explains carefully below, along with specific suggestions for revision. Please revise and respond carefully to these thoughtful comments.

*Reviewer #1 (Recommendations for the authors):*

The authors have addressed my previous concerns. I have no more questions.

*Reviewer #2 (Recommendations for the authors):*

The authors have addressed some of my comments, but not the others. In particular, they should have a more rigorous control of the effect of repetition time, and come up with a better theoretical framework.

Regarding comment #1, the authors did additional analyses to examine the effect of learning efficiency (as measured by the No. of repetitions) and memory. They found no significant linear correlations, and the relationship between semantic relatedness and RF/RI effect remained significant after controlling the learning efficiency.

According to the recursive reminder hypothesis the authors are trying to argue, each time new learning would reactivate the old association, which then affects the strength of old memory and its relationship with the new memory. The number of repetitions thus should have some effect, although it might not necessarily be in a linear fashion. We all know that repetitions do not linearly increase memory strength. Still, this effect should also be modulated by semantic relatedness, which determines the degree of reactivation of old memory. That said, I am not sure if there is a simple statistic method to control the effect of repetition time.

In addition, this confounding factor should also be considered when examining the effect of semantic relatedness on memory interdependency. It should also be considered when comparing the restudy and test conditions, which I will describe below.

Regarding comment #2, the authors acknowledged that the yoke between the repetition time between restudy and test was not perfect due to individual differences, but argued that this was sufficient to match the two conditions. To make this argument, I think the authors should show that the number of repetitions did not affect the overall results for both the test and restudy conditions. They have two choices to prove this. The first option is to randomly assign the number of repetitions to each pair and each subject, or use the group averaged repetitions for each pair (to account for the effect of semantic relatedness on subsequent learning) on every subject. The second option is to use a fixed number of repetitions for all subjects and pairs. I think the second option would be better, considering my comment #1.

Regarding my comment #3, the authors strongly argued that memorability and interdependence should theoretically be dissociated, with which I agree entirely. Nevertheless, with higher memorability, the chance of both items being remembered was also higher. I think a straightforward way to convince the readers is to calculate the correlation between memorability and interdependence and use memorability as a covariate while examining the result of interdependence.

For the Δboth condition, the authors found that Δcue and Δtarget relatedness have a different effect on interdependency and memorability. I agree with the authors that this result did complicate the result and the interpretation. Nevertheless, I think the result is interesting thus should be included and briefly discussed.

My last comment concerns the theoretical framework, which the first reviewer has also raised. In this revision, the authors briefly mentioned the recursive reminder hypothesis in the introduction but did not describe the hypothesis or the predictions. This introduces extra confusion about the relationship between the recursive reminder hypothesis and Osgood's predictions. What are the mechanisms underlying Osgood's predictions? Would the recursive reminder hypothesis support Osgood's predictions?

Moreover, both the recursive reminder hypothesis and Osgood's predictions deal with semantic relatedness. It is thus surprising that at the end of the Introduction, they were to examine several effects, such as the broad and narrow semantic association, the delay effect, the testing effect, and the effect of semantic relatedness on memory interdependency. Would the inclusion of these conditions help to falsify the hypothesis or predictions? The authors examined the semantic relatedness in different conditions separately but did not directly compare these conditions. They argued that this was because they were from different experiments. I am not sure if this argument is valid.

The authors introduce the testing effect to explain the delay effect and the difference between restudy and testing conditions. The testing effect, in my view, is mainly an observation, not an interpretation. Many possible mechanisms have been proposed and they are still under debate.

That said, I should note that this study examined a broad range of essential issues in RF and RI effects, and it is thus a challenge to come up with a simple theoretic framework to cover all of them. Meanwhile, I still think this study has provided critical empirical data for understanding the RF/RI effect. It is also laudable that they make the data open so that other researchers interested in this question could further examine it.

*Reviewer #3 (Recommendations for the authors):*

The authors have thoughtfully responded to the comments I and other reviewers raised. I believe the revised manuscript is stronger and clearer. In particular, I think the theoretical framework is better established and the handling of the recursive reminders account is improved. This is a strong paper that has the potential to be of high impact.

---

## [Author Response]

Essential revisions:1) Provide a clear, coherent theoretical framework to clarify the design motivation, results, interpretation, and conceptual advances being made. While all reviewers were highly positive about the empirical rigor, they all had reservations about the theoretical framing. R1 and R2 commented on the confusion of the multiple theoretical accounts throughout and that it is not clear how your results support or refute or distinguish among theoretical alternatives; R2 raised specific, excellent point about the problem of recursive reminder to explain some data points; R3 also commented the lack of significant conceptual advances and the limited influence outside human episodic memory. The paper needs to be substantially revised to improve on the theoretical framework to address these major concerns.

In response to these thoughtful criticisms, we have extensively revised the introduction to provide a clearer theoretical framework with opposing predictions that we refer back to when interpreting our results throughout the paper. Rather than directly motivating our study from Osgood’s (1949) conjecture about semantic relatedness, we now trace the theoretical motivation for this conjecture back to a still-outstanding question about different mechanisms that may give rise to retroactive facilitation or inhibition. On one hand, we may expect semantic relatedness to primarily increase competition between episodic memories, leading to interference across the board. This account has some empirical support in other paradigms. On the other hand, recursive reminders predict facilitation, if related cues trigger rehearsal of prior memories. We suggest that the patterns observed here (and Osgood’s predictions) support this latter account.

2) The motivation of the specific, central, manipulation choices (narrow vs. wide; measuring semantic relatedness using "strength of association") are currently unclear. Addressing this point may rely on having a clear theoretical framework and may also help address the concern of limited influence raised in Point 1 above. That is, at least those who are interested in the dynamics of semantic memory and the interface between semantic and episodic memory (or learning in general) may also find your results relevant once the semantic manipulations are clearly explained.

We have now provided more information about these design choices in the introduction and results of the paper. In particular, we have clarified that the condition presenting a wider range of semantic relatedness (from highly similar to completely unrelated) was introduced to address concerns that our original condition only included items that were highly similar in absolute terms and therefore did not have sufficient coverage over the full spectrum of relatedness to observe what happens when relatedness approaches the limit of unrelated items. We also now discuss at greater length the tradeoffs of different measures of semantic relatedness.

3) Some methodological details need to be addressed by additional analyses and/or clarification: the confound of the repetition time differences, how yoking subjects to the same learning orders were done, and how dependence scores were computed.

We have clarified these additional concepts and analyses more thoroughly in the appropriate places, and we address them more thoroughly in response to reviewers below. In response to the first point, we have included additional partial correlation analyses that we believe rules out this alternative interpretation of our results. In response to the latter points, we have expanded our explanations of these concepts.

Reviewer #1 (Recommendations for the authors):The study tackles the memory dynamics by behaviorally examining what happens to the old (associative) memory when new (associative) memories are formed, and how it is affected by semantic relatedness (prior, long term, semantic memory). Through a grand experiment with 1000 participants and systematic manipulations of semantic relatedness across cue and target, broader semantic context, and delay between old and new associative learning, they provided a rich and robust empirical dataset showing how semantic relatedness between old and new learning in general strengthen the old memories and the old-new relatedness. A cognitive framework memory dynamic is presented to account for the data.Clearly depicting the behavioral profiles of the dynamic relationships among long term (semantic/associative) memory and the new memory is much desired for the understanding of the complex memory formation and storage. The careful experimental manipulation, the impressive sample size (both subject and stimuli) and the data analyses and visualization, are laudable. The consideration of multiple types of semantic distance measures (internal vs. external) also helps establish the robustness of the effect.

We thank the reviewer for this accurate and kind evaluation.

The value of the empirical results aside, I have concerns about several aspects of the current manuscript.The paper is hard to read, not only because of the richness of the conditions, but mostly because of the lack of a clear and coherent theoretical framework to guide the design and interpretation. The Introduction states that it tests the classical Osgood proposal and predictions were made on that model. But no clear alternatives are analyzed, and it is difficult to appreciate the implications of the results consistent or inconsistent with the proposal. The discussion presented several additional accounts/explanations (e.g., recursive reminders, neural models), but they do not seem to constituent explanations but rather descriptions of the data. What is the relationship between these proposals? What specific assumptions about the memory system is supported/refuted by the data? More specifically, while the different effects of cue- and target- relatedness, semantic context (wide/narrow), and temporal delay are certainly intriguing, they are not clearly motivated or explained. Thus, how exactly memory models are constrained by these empirical findings are not easily appreciated.

We appreciate the constructive criticism about readability. We now further explain the conditions and alternative theoretical frameworks at play earlier in the Introduction rather than in the Results or Discussion. We also take the opportunity to expand the paper in parts to unpack some of our densest ideas and sentences.

We now bring up the most salient opposing theoretical accounts in the Introduction (p. 3):

“Here, we evaluate an over 70-year-old proposal by Osgood (1) that this relationship depends on *semantic relatedness*. In building up to this proposal, we will consider three broad possibilities. The first possibility is that relatedness has no effect on episodic memory. A second possibility is that relatedness across experiences introduces interference between memories. Third, relatedness could trigger reminders of prior information during new learning, causing the rehearsal and strengthening of prior memories.

In order for the first (null) account to be correct, memories must be formed distinctly, and processes operating during both encoding and retrieval must be able to accurately pinpoint and isolate memories without any residual effects or dependence on semantics. We consider this account helpful to consider because these processes clearly *are* affected by semantic relatedness, as many decades of research have shown. For instance, semantic relatedness improves memory when both items of a paired associate are related (2–4), it provides an organizational scaffold for clustering responses during free recall (5–8), and it can create false memories for highly related associate words (9, 10). In favor of the second (interference) account, pairing a single retrieval cue with multiple target responses could benefit from greater semantic differences between the targets, allowing for easier dissociation between them (11). Indeed, increasing relatedness between tasks in some paradigms can increase interference (12–14) and/or the rate of intruding material from one task to the other (15–19). Finally, in favor of the third (strengthening) account, there is evidence that we are not always passive during new learning: sometimes we “think back” to, and thereby reactivate, prior experiences (20). These events, called recursive reminders, can occur when subjects are given explicit instructions or cues as reminders (21–23), or – more relevantly here – they can occur spontaneously when information is related (24, 25). Moreover, recursive reminders seem to create interdependence between old and new information, with preserved information about the temporal order of learning rather than source confusion and negative competition between the traces (20, 26–28). The recursive reminders account therefore predicts that semantic relatedness would promote RF and interdependence among memory traces. Altogether, the first account is clearly incorrect, but when and how strongly the countervailing forces of RI and RF from the latter accounts operate remains a central puzzle.”

Following this, we refer back to these countervailing processes throughout the paper, such as in the Results (p. 11):

“We also asked whether target relatedness would increase intrusions, or errors from the secondary pair list into the base pair list. That is, we wanted to contrast two accounts. Under an RI-based account, the targets may merge or compete, leading to confusion about the list contexts (e.g., peace-razorshave). Theoretically, this account could produce some intrusions in addition to RF; indeed, lack of interference in RI studies wherein targets are related has been posited to stem from a “loss of differentiation” between semantically related sources (15), and other studies have found greater intrusion errors with increasing relatedness (16–19). Under the recursive reminders account, highly related new targets would simultaneously strengthen old memories due to reminders of the base pair list and would be scaffolded to the cue as part of the secondary pair list, meaning the list contexts remained interdependent, yet distinguishable (e.g., peace-razor-base list/peace-shave-secondary list). We therefore asked whether relatedness increased across-list intrusions of the new target response into the base pair list by correlating the across-subject intrusion rate with target relatedness. In fact, intrusions significantly decreased in the wider stimulus set, 5-min experiment (*p* < 0.001) and otherwise did not increase with target relatedness in any experiment (all *p* > 0.08; Figure 3-Supp 1), supporting the recursive reminders account.”

and in the Discussion (p. 21):

“Critically, the absence of greater intrusions with higher relatedness suggests that temporal order information regarding the learning list of each pair was preserved rather than the two contexts becoming confused. That is, with high relatedness, recursive reminder-induced memory benefits outweighed the possible countervailing force of competition at retrieval. With weaker pre-existing associations, we speculate that subjects noticed changes while learning new secondary pairs but may have been less likely to recall and integrate them with corresponding base pairs.”

We have now added motivation regarding the two retention intervals to the Introduction (p. 5):

“Additionally, interference often differs depending on the delay between learning interfering material and test (23, 51–55), and we therefore fully crossed the narrower and wider stimulus sets with two different test delays occurring 5-min and 48-hr after secondary pair learning (Figure 1C).”

Regarding the two ranges of relatedness: our initial set of experiments featured the stimulus set with a narrower range of relatedness. As detailed in the paper, we initially found intriguing results regarding the influence of relatedness on memory, but we did not find any semblance of retroactive interference. We reasoned that perhaps a wider range of relatedness would both generalize the results more broadly beyond local semantic neighbors of direct associations and also potentially demonstrate that retroactive interference still occurs in this paradigm (which we indeed found under low relatedness in the 5-minute delay experiment). We also have now added more motivation regarding the two ranges of relatedness to the Introduction (p. 5):

“In our initial experiments, we used a stimulus set with a narrow range of relatedness values, corresponding to the direct associative pair strength. In later experiments, to address how these initial results generalized beyond local semantic neighborhoods of direct associations, we used a stimulus set with a wider range of relatedness that included truly unrelated associations.”

A further concern regards the treatment of "semantic effects/semantic relations". The study distinguishes old and new memory, the "semantic effects" they observed are part of an "older" memory. The way the authors use "semantic relatedness" to describe the effect of "prior knowledge" does not make distinctions between potentially different types of long-term relations in the memory representations such as semantic vs. associative -- in the main analyses the semantic distance were operationalized using association strength. It's okay to be parsimonious and show that the distinction is not necessary, and they affect memory dynamics in the same way. But this needs to be explicitly discussed, given the psycholinguistic literature in the past decades that these two kinds of relations show different effects in comprehension (priming) and production (e.g., picture-word interference) tasks. More generally, the mechanisms underlying associative and semantic memory/learning, and between episodic and semantic memory are expected to be discussed in greater depth and clarity (e.g., see Renoult et a., 2019 TICS). It is also not clear what the psychological reality of one step or multiple steps of semantic relations.

We thank the reviewer for raising this point. Indeed, semantic and associative relations are dissociable and affect cognition in different ways (e.g., Thompson-Schill et al., 1998). We have not made it an explicit point to dissociate them in this study. However, as the reviewer notes, our analyses detailed in Supp File 6 do not show clear differences between measures of semantic and contextual similarity (e.g. using LSA and word2vec) versus associative strength (e.g. based on free recall networks). We have added the following as a limitation to the discussion (p. 25):

“A notable limitation of our study is that we did not dissociate between semantic representations based on distributed representational models [e.g., (61, 131)] and associative relations based on association strength and spreading activation [e.g., (132)], instead relying on the broad term, semantic relatedness, to capture both constructs. Analyses in Supp File 6 did not indicate any clear distinction in correlations with memorability and interdependence between measures of semantic and contextual similarity such as LSA and word2vec versus associative strength. Nevertheless, we did not aim to directly contrast these two constructs by selecting stimuli that dissociate them, and given that the two types of relations show different effects on cognition [e.g., (133)], this will be important in future work.”

The reviewer refers to the literature on interactions between episodic and semantic memories as well as the blurry boundary between them (Renoult et al., 2019). We agree with the general sentiment in this literature that the distinction is impure and that subjects likely use mental strategies involving semantic memory here. In addition to other ways this is mentioned in the Discussion section, we have added the following second sentence (first sentence included for context) (p. 22):

“In this framework, our high relatedness ΔTarget and ΔBoth conditions resemble a closed loop, except that rather than having to learn the final link in the loop de novo, unrelated associations are closed by the pre-existing target relationship in the ΔTarget condition and by both pre-existing cue and target relationships in the ΔBoth condition. These results demonstrate a clear interplay between semantic and episodic representations, whereby semantic representations scaffold the formation and retention of episodic memories (8, 104).”

The psychological reality of one versus multi-step relationships is an interesting topic that has been addressed in at least a few studies. Nelson et al. (1997), Nelson & Zhang (2000), and Kenett et al. (2017) showed that recall can be facilitated by two-step relationships, while De Deyne et al. (2019) showed that multi-step relationships could successfully predict human similarity in a word association game. These findings are mentioned in the Results (p. 16):

“Additionally, semantic network relationships can predict paired associate memory beyond single steps to nearby neighbors, with significant benefits shown up to two (3, 64) or three (65) semantic steps.”

In summary, I appreciate the empirical rigor of the study and find the results potentially very important. But they need to be situated in clearer theoretical framework to show their implications for how memory works and to guide further cognitive/neural studies of memory.L69 ". In studies in which new and old cues and targets were 70 semantically related, or new cues were related to old cues while new targets were not related to 71 old targets, RI for the original A-B association has been found on the order of minutes (31-34). 72 However, to our knowledge, no study has investigated long-term effects when both new and old 73 cues and targets are related, or with very high levels of semantic relatedness."- I find this sentence very hard to follow.

We thank the reviewer for pointing this out. The sentence now reads (p. 4):

“Studies in which both cues and targets bear some level of relationship to the original A-B pair are scant. However, there have been cases where *either* the new cue was semantically related to the old cue but the targets were unrelated, the new target was semantically related to the old target but the cues were unrelated, or both new cues and targets shared some modest level of relatedness with the old ones; in each of these cases, RI for the original A-B association has been observed when testing occurred after short retention intervals (on the order of minutes) (13, 46–48). However, to our knowledge, no study has investigated longer-term memory in cases where the new cues and targets were both highly related to the old ones.”

Reviewer #2 (Recommendations for the authors):This study has several strengths. It did a comprehensive, vigorous hypothesis test, by examining multiple experimental conditions under the same paradigm and with a large sample size. These results could help to reconcile the mixed findings in the literature. Their treatment of semantic relatedness is also thorough and appropriate. The conclusion is mostly accurate and justified.

We thank the reviewer for their kind assessment.

There are some methodological details that I think should be considered. Finally, the lack of neural evidence somewhat limits its conceptual advances.My main suggestions are related to the methodological details, which I will detail below.First, the current study used a retrieval-to-criterion learning paradigm, which is very efficient in achieving the desired learning outcome. Although I think this is appropriate for the initial learning of base pairs, it might introduce additional variances during the learning of second pairs that may complicate the explanations. As depicted in Figure S7, semantic relatedness facelifted new learning, resulting in fewer repetitions for strongly related pairs than weakly related pairs. Since this new learning also influences old memory, it is unclear whether the different repetitions also matter, in addition to semantic relatedness itself.

The reviewer makes an astute point that the efficiency of new learning in the secondary pair condition could theoretically predict retroactive memory benefits. We address this concern with new analyses and the following text in the Results section (p. 19):

“Next, we wanted to rule out an alternative possibility raised by these results. Secondary pairs with high relatedness were learned more efficiently, meaning that they had fewer exposures. If the number of exposures increased RI, this would suggest our RF effects could stem in part from lesser interference. We conducted two analyses to address this possibility. First, we correlated new learning efficiency with memorability across pairs in each condition. We found generally weak evidence in favor of this idea, with significant (*p* < 0.05) results in only the ΔTarget condition in the wider stimulus set, 48-hr delay experiment (*r* = 0.30, *p* = 0.02). Second, we ran partial correlations between relatedness and memorability across pairs while controlling for new learning efficiency. These partial correlations remained significant in all of the main analyses above, including in the ΔTarget condition in the narrower stimulus set, 48-hr experiment (*r* = 0.34, *p* = 0.026), wider stimulus set, 5-min delay experiment (*r* = 0.45, *p* = 0.002), and wider stimulus set, 48-hr delay experiment (*r* = 0.36, *p* = 0.016) and for cue+target relatedness in the ΔBoth condition in the narrower stimulus set, 48-hr experiment (*r* = 0.41, *p* = 0.005). Full results from these partial correlations can also be seen in Supp File 7. Therefore, it appears our RF effects did not rely on the amount of pair exposure during secondary pair learning.”

Given the interest in this point and in other reviewer comments, we also now feature these learning results more prominently as Figure 8 in the paper.

Second, the comparison between restudy and retrieval practice is very interesting and informative. The authors also made a good effort to match the exposure, by yoking each of 200 subjects to the exact learning order of subjects in the narrower stimulus set, 48-hr delay experiment. However, it is not clear how this was achieved, given the individual differences in the overall memory performance. This could be more complicated considering the memorability of a given pair also differs across subjects.

We thank the reviewer for the opportunity to clarify this point. We have now added the following to the Methods section (p. 28):

“In the study-only experiment, the yoking procedure for each subject was matched to the learning order of a subject from the retrieval-to-criterion, narrower stimulus set, 48-hr delay experiment. For example, subject #1 from the retrieval-to-criterion experiment was matched with subject #1 from the study-only experiment. We copied the learning order precisely in both base and secondary pair learning phases from each retrieval-to-criterion subject to the study-only subject, so if subject #1 in the former saw ‘sick-push’ first, so did subject #1 in the latter experiment, and so on. In later rounds of learning, when many word pairs had dropped out for a subject in the retrieval-to-criterion experiment, those same items dropped out for the yoked subject in the study-only experiment. We acknowledge that this procedure cannot control for individual differences in memorability; for instance, subject #1 in the retrieval-to-criterion experiment and subject #1 in the study-only experiment could require different exposures to achieve the same criterion, so the pairs remaining at the end of each learning phase may not be the ones the study-only subject would have struggled to learn. Such differences cannot be addressed without an additional test assessment in the study-only experiment, but we believe that matching the objective amount of exposure serves as a sufficient control for our purposes.”

We also thank the reviewer for their enthusiasm regarding the restudy (study-only) experiment, and we have now made those results more prominent in the paper by including them as Figure 7.

Third, the examination of memory dependence is also very interesting. I am curious how the dependence score is dissociated with the memory score? Still, one can imagine that this index might be less meaningful for subjects who performed very badly or very well in both tests, as their high dependence score might not reflect the true memory integration.

We thank the reviewer for this important point. Memory recall and dependence are indeed dissociable measures. We calculated dependence by considering, for a given ‘duo’ within a particular condition (e.g., base pair memory for ‘razor’ and then secondary memory for ‘shave’, given the pairs ‘peace-razor’ and ‘peace-shave’, in the ΔTarget condition), how often the duo was either both remembered, or both forgotten. So, a duo could potentially have a higher rate of dependence than raw memory performance if duos were also regularly forgotten together (e.g., dependence could be 0.9 if they ‘moved together’ for 90% of participants, even if the raw base pair memory recall rate was 0.7 and the secondary pair rate 0.6). Conversely, the duo could have a lower rate of dependence than memory performance if one of the two pairs are regularly recalled and the other not. Therefore, the measures are dissociable. To assess whether dependence could be high simply because of overall ceiling (or floor) performance, we also calculated, as a baseline, how often ‘razor’ was remembered against all other words in the secondary condition. This was plotted as the dotted line in each of the dependence graphs. Therefore, while we agree that integration performance is difficult to assess for individuals with extremely high or extremely low performance, we would argue that such subjects would simply add noise to our measure, and we have many subjects between these extremes. We have clarified these points by modifying the Methods section (p. 28-29):

“Memory dependence was calculated by first examining memory for each base pair target-secondary pair target duo within a condition across subjects. Consider the following example in the ΔTarget condition, where 1 = correct and 0 = incorrect memory and values are represented respectively across subjects. If answers for subjects #2, 7, 12, and 17 when given “peace” as the cue during the base pair test (correct answer: “razor”) were 1,1,1,0 and their answers when given “peace” as the cue during the secondary pair test (correct answer: “shave”) were 1,1,0,0, the across-subject dependence would be 0.75. Conversely, if the base pair test across the same subjects for the same pair was 1,1,1,0 and the secondary test was 0,1,0,1, the across-subject correlation would be 0.25. It is important to note that, between the extremes of ceiling and floor performance, dependence and performance are dissociable. A target duo could potentially have a higher rate of dependence than raw memory performance if it regularly became forgotten together. Conversely, a target duo could have a lower rate of dependence than memory performance if one of the two pairs are regularly recalled and the other not.”

Fourth, the motivation of some experimental manipulations needs more justifications. For example, why did the author manipulate a narrower vs. a wider range of semantic relatedness? For the restudy vs. retrieval comparison, why the narrower stimulus set, 48-hr delay condition was selected?

[Regarding the range of relatedness, we have copied part of this justification from our response to a similar inquiry to R1 above.] Our initial set of experiments featured the stimulus set with a narrower range of relatedness. As detailed in the paper, we initially found intriguing results regarding the influence of relatedness on memory, but we did not find any semblance of retroactive interference. We reasoned that perhaps a wider range of relatedness would both generalize the results more broadly beyond local semantic neighbors of direct associations and also potentially demonstrate that retroactive interference still occurs in this paradigm (which we indeed found under low relatedness in the 5-minute delay experiment). We have now clarified this in the Introduction (p. 5):

“In our initial experiments, we used a stimulus set with a narrow range of relatedness values, corresponding to the direct associative pair strength. In later experiments, to address how these initial results generalized beyond local semantic neighborhoods of direct associations, we used a stimulus set with a wider range of relatedness that included truly unrelated associations.”

Regarding the restudy vs. retrieval experiment, this is a worthy question of why we selected the narrower stimulus set, 48-hr delay condition. While the wider range stimulus set could have potentially allowed us to show semantic relatedness effects across a broader range of the semantic space, the narrower set offered the opportunity to test whether the benefits we demonstrated in the ΔBoth condition in the prior, retrieval-to-criterion experiments extended to the study-only condition. If we had used the wider stimulus set and found null results in the ΔBoth condition, it would have been unclear whether it was because of the stimulus set or learning strategy. We have clarified this point in the Results section (p. 17):

“We chose this stimulus set and delay because we were especially interested if the results in the ΔBoth experiment from the otherwise equivalent retrieval-to-criterion experiment would generalize to study-only conditions.”

My final comment is related to the theoretic account of the results. It seems that the interpretation of the results primarily relies on the recursive reminders account. Although I agree this is a reasonable choice, there are some issues that are worth mentioning. For example, this account might not explain the results under the ΔCue condition very well.

We thank the reviewer for these insightful points. Regarding the extent to which recursive reminders accounts for the ΔCue condition, we note in the Discussion that the recursive reminders account could be relevant for the ΔCue condition given the weakly positive correlation with relatedness (meaning reminders are more likely as relatedness increases). However, we also note the alternative possibility of increasing the availability of the target item. Even if target availability turns out to capture more of the retroactive benefits in this condition, it does not necessarily negate the relevance of the recursive reminders account for the other conditions – it simply means there are multiple possible memory mechanisms at play. We have changed the Discussion slightly to acknowledge this extra ambiguity as follows (p. 22):

“Altogether, the retroactive benefits are clear in the ΔCue condition, but the mechanism seems to differ from the ΔTarget condition and the extent to which recursive reminders are specifically involved versus other processes such as increased target availability is unclear.”

It is unclear whether these effects were mainly driven by the ΔTarget under the ΔBoth condition.

Contrasting the relatedness of the new target word versus the new cue word within the ΔBoth condition is a fantastic idea. First, we performed additional analyses on memorability in the ΔBoth – control condition that would separately correlate with ΔCue or ΔTarget relatedness in the narrower stimulus set, 48-hr delay experiment (where we observed ΔBoth RF). As a reminder, correlating memorability against the summed ΔCue + ΔTarget relatedness value produced a significant correlation (*r* = 0.40, *p* = 0.007). Looking separately, we found that ΔTarget relatedness correlated with ΔBoth memorability (*r* = 0.38, *p* = 0.01), whereas ΔCue relatedness did not (*r* = 0.17, *p* = 0.26). Furthermore, the ΔTarget relatedness correlation survived significance when performing partial correlations controlling for ΔCue relatedness (*r* = 0.39, *p* = 0.009).

We next performed the same analyses on base pair-secondary dependence in the ΔBoth condition. As a reminder, correlating dependence against the summed ΔCue + ΔTarget relatedness value produced a significant correlation (*r* = 0.31, *p* = 0.04). Looking separately, we found that ΔCue relatedness correlated with ΔBoth dependence (*r* = 0.30, *p* = 0.04), whereas ΔTarget relatedness did not (*r* = 0.12, *p* = 0.42). Furthermore, the ΔCue relatedness correlation survived significance when performing partial correlations controlling for ΔTarget relatedness (*r* = 0.31, *p* = 0.04).

These are potentially important effects. However, there is inconsistent importance of cue versus target for dependence and memorability correlations, respectively, and there is (to us) no obvious, clear connection between these differing findings and those in the ΔCue and ΔTarget conditions. Therefore, we believe including them in the manuscript would risk further complicating the paper and also risk our over-interpreting these results before this unclear relationship is replicated. We propose retaining the previous analyses using bivariate cue/target relatedness and summed cue+target relatedness, which require fewer assumptions. While they may be underspecified, we do not believe they are incorrect. Nevertheless, we appreciate that there could be interesting nuances here, and we would be happy to include these analyses if the reviewer disagrees. Finally, we note that the data and code have been released and can be re-analyzed if this becomes a focus point of future investigations. As a result of this ambiguity about what exactly is driving the ΔBoth effects, we have softened the language that cue and target relatedness must be both high in the Discussion section, of which the new text reads as follows (p. 22):

“Strikingly, in the ΔBoth condition, RF occurred overall (versus control) and at high values of cue and target relatedness in the narrower stimulus set, 48-hr delay experiment. Moreover, base pair-secondary pair dependence similarly occurred at high cue and target relatedness in this experiment. These results suggest that, as in the ΔTarget condition, there was an increased likelihood of recursive reminders supported by pre-existing associations. These effects occurred when cue and target relatedness were high, possibly because changing both causes subjects to fail to recollect both pre-existing links and/or attribute the secondary pair to a new memory (103).”

It is also not readily transparent to me how it could account for the effect of study-test interval on the overall RI vs. RF patterns. Still, the authors did not discuss how it could account for the restudy vs. retrieval differences.

Regarding how recursive reminding could have different effects at different retention intervals: if recursive reminders act as a form of memory retrieval, this part of the effect could function like testing effects (Karpicke & Roediger, 2008), which have preferential benefits at longer delays. Therefore, as recursive reminders become more likely, the benefits (relative to the control condition) should be more reliable after 48 hours than immediately, and this supports our general pattern of results. This explanation would also account for the long-term benefits of retrieval relative to re-study. (We do not directly compare them in the paper because they were part of different experiments, but this trend is readily apparent in examining the results and the general finding is highly consistent with the literature). To clarify this point, we have added the following to the Discussion section (p. 21):

“Although we have largely avoided comparing 5-min and 48-hr delay results directly because they were part of different experiments, it appears visually that conditions in which we expect recursive reminders tend to slow the rate of forgetting (relative to the amount of forgetting in the control condition). This pattern suggests recursive reminders act as effective retrieval processes, similar to the testing effect (20).”

Moreover, since there is no neural evidence to actually examine the reactivation during learning, it is my opinion that the current study did not provide direct evidence to support or refute this account, which somehow limits the conceptual advances of the current study. For the same reason, I am not sure if the neural model proposed in the discussion (Figure 7) is justified by the data.

Regarding neural evidence and the neural model (now in Figure 9) – we acknowledge that we do not have direct evidence showing reactivation of prior memory traces. It has been shown numerous times that reactivation of old information during new learning occurs and supports memory for the old information (e.g., Kuhl et al., 2010; Chanales et al., 2019), and we effectively built upon those findings to discuss how our effects may arise neurally. We do believe the integration of semantic relatedness and recursive reminders into a unified model has some benefit for the literature. However, this model is still speculative, and we further acknowledge this point in the Discussion section (p. 23):

“Given these findings, we now speculate on how our effects fit within a neurobiological framework extending the recursive reminders account (Figure 9). In our study, pre-existing semantic relationships existed primarily within the neocortex, while previously unrelated pairs were bound by the hippocampus along with their episodic list context and novel, related episodes could also have been interdependently linked within the hippocampus. The No Δ condition generally produced maximum strengthening for both base and secondary pairs. In line with a theory suggesting that even repetitions of the same learning material creates multiple traces within the hippocampus (115), these context memories were formed individually yet, as shown by their strong interdependence (red, leftmost point in Figure 6B), they were strongly bound within the hippocampus. In the ΔTarget and ΔBoth conditions (and to a lesser extent in the ΔCue condition), high relatedness between base and secondary pairs – represented by pre-existing cortical links – facilitated base pair reactivation via recursive reminders, thus creating an inter-hippocampal association and behavioral dependence. Such benefits were far more limited with no or low relatedness. Additionally, though we propose that these interdependencies initially rely on the hippocampus, the novel associations may eventually become linked within the cortex with more repetitions (57, 105). Altogether, the model in Figure 9 provides a simplified, speculative explanation for our collected results and a testable mechanism for future studies.”

Although this study used a large sample, some of the effects seem to be unreliable. For example, in figure S8, when both the narrow and wide conditions were merged, the effect of semantic relatedness show different trends, i.e., the regression line did not overlap.

We acknowledge that the effects are not always the same between the narrower and wider stimulus sets. In some cases (notably, the ΔBoth condition) we attribute to possible differences in overall relatedness and the likelihood of engaging in recursive reminders, such as in the Discussion (p. 22):

“Additionally, null effects in the wider stimulus set experiments – even for pairs with high cue and target relatedness – suggest that relatedness across pairs may also need to be high on average for subjects to start noticing and recollecting changes rather than attributing the pairs to entirely new associations.”

Did the authors also examine the strength of semantic association between the two words in each pair, and whether this could be a confounding factor that should be controlled in the analyses?

We thank the reviewer for this point. We note that we specifically designed the experiment to rule out these types of confounds. That is, the memorability analysis allows us to subtract memory from the *same exact pairs* between one condition and the other, indicating that any incidental relationships due to the stimuli would not affect our results. Nevertheless, given that semantic relationships often benefit word pair learning and memory in countless other studies [e.g., Lyon, 1914; Noble, 1952; Underwood & Schultz, 1960; Bahrick, 1970; Hall, 1972; Nelson et al., 1992; Naveh-Benjamin, 2000; Payne et al., 2012], we explicitly performed these analyses by investigating memory in the control condition versus the GloVe values between cues and targets. Here we did not find any significantly positive relationships, and in one case, we found a negative relationship [narrower set, 5-min experiment: *r* = 0.17, *p* = 0.14; narrower set, 48-hr experiment: *r* = 0.07, *p* = 0.37; wider set, 5-min experiment: *r* = -0.41, *p* = 0.003; wider set, 48-hr experiment: *r* = 0.05, *p* = 0.30; study-only experiment: *r* = 0.15, *p* = 0.79]. It is unclear why the negative relationship emerged in only one study, but we highlight that these results may differ from the positive relationships in the literature because there was very little meaningful variance along the semantic relatedness dimension between cues and targets. That is, cues and targets were *intentionally* unrelated, so most of the variance in GloVe values was restricted to the low range of possible values (the mean±stdev GloVe value for combined stimulus sets was 0.19±0.11 on a scale that could be as high as 0.95). This was by design, as we intended the relevant semantic relationships in the study to be between old and new cues and old and new targets rather than between cues and targets themselves. Furthermore, we note that because our main memorability analyses involved contrasting memory for each pair in the experimental condition with the same pair in the control condition, we account for the influence of incidental differences in cue-target semantic relationships. Here again we opt not to include this in the paper to avoid complicating the message, but we could do so if the reviewer disagrees.

Reviewer #3 (Recommendations for the authors):Overall, this is a very solid and rigorous paper. The main contribution is to understanding the factors that determine behavioral measures of retroactive interference versus facilitation. In particular, when cue-target associations change (e.g., A-B to A-D), the question is how the degree of similarity between original and new cues and original and old targets will influence the relative interference vs. facilitation effects. This issue has a very long history in experimental psychology. The paper takes particular inspiration from a very influential idea from Osgood that interference/facilitation effects can be described along a continuous surface where cue similarity and target similarity are independent dimensions. The paper generally supports the ideas proposed by Osgood, but represents what is likely the most thorough empirical test of these ideas. Notably, whereas Osgood's original ideas were inspired by combining findings from many different experiments/papers/researchers, the current study packs all of the key experimental variables into a single, cohesive set of experiments.Other strengths of the paper include relatively large sample sizes, consideration of the effects of delay (5 min vs. 48 hours) and consideration of the effects of retrieval practice vs. study. Additionally, although not a major point of emphasis in the paper, one of the interesting findings is that similarity between original and new items can protect older memories without increasing confusability (intrusions). This perhaps represents the most surprising result in the paper as the alternative prediction would be very reasonable (that semantic relatedness would increase intrusions). It seems difficult to pin down exactly why relatedness was protective against intrusions, but it is an intriguing result.The biggest limitation in the paper is the amount of conceptual advance. As the authors note, Osgood's original ideas anticipated (most of) the key results in the current study-namely, the idea that facilitation and interference are a function of the similarity between original and new information. While the current paper provides a much more definitive investigation of this than is currently available in the literature, there have also been many papers since Osgood (nicely reviewed by the authors) that anticipate aspects of the current results. In particular, studies looking at integration or recursive reminders have clearly established that processes following initial encoding (of the base pairs) will impact the degree of interference and can even promote facilitation. In the current study, there is no direct evidence that integration or reminders occurred, but it is inferred that relatedness triggered reminders.Taken together, the paper is likely to be of interest to those that study episodic memory in humans (and, in particular, those interested in memory interference). Given the influence of Osgood's original paper in the field of human learning and memory, this paper may also have substantial influence as a definitive test of Osgood's ideas. However, it is less clear whether the paper will have substantial influence outside this field.

We thank Dr. Kuhl for his kind words. We took note to try to clarify the advance in our revised version in response to this critique and the critiques offered by the other reviewers. Due to his and the other reviewers’ suggestions, we have also set up the alternative prediction of increased intrusions / confusability as a major alternative account in the Introduction and throughout the paper.

[Editors' note: further revisions were suggested prior to acceptance, as described below.]

Reviewer #2 (Recommendations for the authors):The authors have addressed some of my comments, but not the others. In particular, they should have a more rigorous control of the effect of repetition time, and come up with a better theoretical framework.Regarding comment #1, the authors did additional analyses to examine the effect of learning efficiency (as measured by the No. of repetitions) and memory. They found no significant linear correlations, and the relationship between semantic relatedness and RF/RI effect remained significant after controlling the learning efficiency.According to the recursive reminder hypothesis the authors are trying to argue, each time new learning would reactivate the old association, which then affects the strength of old memory and its relationship with the new memory. The number of repetitions thus should have some effect, although it might not necessarily be in a linear fashion. We all know that repetitions do not linearly increase memory strength. Still, this effect should also be modulated by semantic relatedness, which determines the degree of reactivation of old memory. That said, I am not sure if there is a simple statistic method to control the effect of repetition time.

We thank the reviewer for questioning why we did not find a repetition effect. We agree in theory with the reviewer's logic that additional repetitions should retroactively benefit memory under a recursive reminder account, if participants thought back to an equal extent on each trial. However, we believe there are other possible factors at play that complicate the relationship.

First, successful reactivation might improve the likelihood of success on the current trial, leading that word pair to be repeated less in the future (given the adaptive design). Since greater relatedness leads pairs to be learned more efficiently, there may be approximate parity between a small number of highly effective recursive reminders (i.e. high relatedness pairs) and a greater number of less effective recursive reminders (i.e. low relatedness pairs) that required more repetitions. In other words, all pairs could eventually end up with an effectively similar number of recursive reminders over the course of the experiment.

Second, thinking back could cause momentary blocking of the current (secondary pair target) on some trials, perhaps making the subject think back to the prior phase less (or even mentally suppress the memory) on subsequent repetitions. There are likely differences in these factors across individuals and even across trials within the same individual.

This is a fascinating set of hypotheses, but we believe that resolving the dynamics of how and when subjects may think back, depending on current goals and current trial success requires a more targeted effort (e.g. explicitly manipulating the number of repetitions, as the reviewer suggests below) that falls beyond the scope of this paper. We believe our new analysis — a partial correlation that controls for learning efficiency —- is sufficient for the present claim: that the existence of retroactive memorability benefits does not necessarily rely on learning efficiency during secondary pair learning.

In addition, this confounding factor should also be considered when examining the effect of semantic relatedness on memory interdependency. It should also be considered when comparing the restudy and test conditions, which I will describe below.

To account for possible confounds of learning efficiency in our interdependence analysis, we have now calculated (a) correlations between efficiency and interdependence and (b) partial correlations between semantic relatedness and interdependence while covarying out efficiency. These forms the new supplementary file 10.

Regarding (a), correlations between learning efficiency and interdependence were negative (just as the ones we previously observed between efficiency and semantic relatedness).

Regarding (b), the partial correlations between relatedness and interdependence survive in the ΔTarget condition and in the retrieval-to-criterion, wider stimulus set, 48-hr experiment in the ΔCue condition. In the case of the ΔBoth condition in the retrieval-to-criterion, narrow stimulus set, 48-hr experiment, the correlation was now only marginally significant. Our findings are largely robust to controlling for secondary pair learning efficiency, so it is unlikely that it can account for our findings.

Regarding comment #2, the authors acknowledged that the yoke between the repetition time between restudy and test was not perfect due to individual differences, but argued that this was sufficient to match the two conditions. To make this argument, I think the authors should show that the number of repetitions did not affect the overall results for both the test and restudy conditions. They have two choices to prove this. The first option is to randomly assign the number of repetitions to each pair and each subject, or use the group averaged repetitions for each pair (to account for the effect of semantic relatedness on subsequent learning) on every subject. The second option is to use a fixed number of repetitions for all subjects and pairs. I think the second option would be better, considering my comment #1.

We thank the reviewer for encouraging us to think more critically about this point. We agree that collecting additional data using one of the suggested designs would provide the strongest evidence for this argument, but we are hesitant to collect more data to iron down this point, especially as the two groups, in keeping with our prior sample sizes, would require 400 subjects and an entire year of sampling our subject pool (since we do not have an Intro Psychology pool in the Spring term).

However, the reviewer’s point is well taken, and we now explicitly acknowledge this limitation. Specifically, we have changed the following passage on p. (28) from:

“Such differences cannot be addressed without an additional test assessment in the study-only experiment, but we believe that matching the objective amount of exposure serves as a sufficient control for our purposes.”

to:

“Such differences cannot be addressed without an additional test assessment in the study-only experiment. We believe that matching the objective amount of exposure was the best way to match restudy to retrieval-to-criterion learning conditions, but another study using a fixed number of repetitions for each pair in restudy and test conditions or a study which randomly assigns the number of repetitions across pairs in restudy and test conditions would address the role of individual differences more precisely.”

Regarding my comment #3, the authors strongly argued that memorability and interdependence should theoretically be dissociated, with which I agree entirely. Nevertheless, with higher memorability, the chance of both items being remembered was also higher.

We appreciate this comment and offer a more extensive argument in our revision. Our claim is that semantic relatedness drives both memorability and dependence, so we believe that the fact that memorability and dependence are both correlated with semantic relatedness is not an issue for the measurement. This relationship is expected if the memorability of the secondary pair (in addition to memorability of the base pair) is higher when relatedness is higher. Base pair memorability does not *necessarily* positively predict secondary pair memorability (and therefore also dependence); in fact, we have another dataset where subjects learn two rounds of unrelated word pairs which have no relationships between target words (i.e., a more classic A-B, A-D task). In this dataset, greater memorability in A-B pairs does not predict greater memorability for the A-D pairs and thereby does not predict greater dependence. (In fact, in that unpublished study, greater A-B memorability *negatively* predicts A-D memorability, in line with classic inhibitory effects like blocking, or a competitive process.)

In this unpublished study, we were more interested in the effects of temporal context and RI/RF. Subjects learned 44 unrelated word pairs (A-B) before learning 22 potentially interfering A-D pairs. (Note that in the terms of the present paper under revision, A-B pair learning = base pair learning and A-D pair learning = secondary pair learning in the ΔTarget learning). To reiterate, there was no semantic relationship between B and D words. A final test was administered 48 hours later for A-B and then A-D pairs. For this unpublished study, subjects were randomly assigned to have A-D learning occur either immediately after A-B learning (5 min), 3 hr after learning, halfway between learning and test (24 hours later), 3 hr before the test, or immediately before test (5 min). The critical finding for the present purposes is that, collapsing across all A-D learning time conditions, A-B memorability in this study was *negatively* (rather than positively) correlated with A-D memorability (*r* = -0.34, *p* = 0.025). One could presumably also create conditions in which the two measures are uncorrelated; the point here is that base pair-secondary pair positive dependence is not simply an artifact of base pair memorability.

I think a straightforward way to convince the readers is to calculate the correlation between memorability and interdependence and use memorability as a covariate while examining the result of interdependence.

Regarding this point, we have calculated correlations between across-word pair memorability and dependence. Additionally, we believe that, in the last sentence, the reviewer meant to suggest that we calculate correlations between *relatedness* and dependence while partialling out base pair memorability, so we have also calculated that. (We do apologize if we misinterpreted this suggestion, and we would be happy to run a different calculation if we have done so!) We have also computed the converse correlations between relatedness and memorability while partialling out dependence. These correlations have been placed in the new Supplementary Files 7 and 8.

For the Δboth condition, the authors found that Δcue and Δtarget relatedness have a different effect on interdependency and memorability. I agree with the authors that this result did complicate the result and the interpretation. Nevertheless, I think the result is interesting thus should be included and briefly discussed.

We thank the reviewer for this feedback. We have now added a short presentation of these relationships to the paper in the Results section (p. 14):

“We next explored whether cue or target relatedness differentially affected memorability and base pair-secondary pair dependence within the ΔBoth condition. We found that target relatedness correlated with ΔBoth memorability (*r* = 0.38, *p* = 0.01), whereas cue relatedness did not (*r* = 0.17, *p* = 0.26). Furthermore, the target relatedness correlation survived significance when performing partial correlations controlling for cue relatedness (*r* = 0.39, *p* = 0.009). Conversely, we found that cue relatedness correlated with base pair-secondary pair dependence in the ΔBoth condition (*r* = 0.30, *p* = 0.04), whereas target relatedness did not (*r* = 0.12, *p* = 0.42), and the cue relatedness correlation survived significance when performing partial correlations controlling for target relatedness (*r* = 0.31, *p* = 0.04). Therefore, although our primary analyses in the ΔBoth condition focused on the bivariate effects of cue and target relatedness, the two measures have dissociable impacts on memorability and dependence.”

We also briefly mention these effects in the Discussion (p. 22):

“We do note that, in the narrower stimulus set, 48-hr delay experiment, cue and target relatedness had somewhat dissociable effects in the ΔBoth condition, such that target relatedness predicted memorability while cue relatedness predicted dependence. We find these results intriguing and worthy of further investigation in future work.”

My last comment concerns the theoretical framework, which the first reviewer has also raised. In this revision, the authors briefly mentioned the recursive reminder hypothesis in the introduction but did not describe the hypothesis or the predictions. This introduces extra confusion about the relationship between the recursive reminder hypothesis and Osgood's predictions. What are the mechanisms underlying Osgood's predictions? Would the recursive reminder hypothesis support Osgood's predictions?

These are excellent questions. First, Osgood’s paper mostly characterized the budding literature (before 1949) and offered no mechanisms by which these effects would arise. This is perhaps surprising, but also perhaps not, as it was published during the behaviorist period before the “cognitive revolution”. (Indeed, even word pair associations were referred to as “stimulus” and “response” in the paper.) Therefore, there were no original mechanisms proposed for *how* these effects would arise, but we do indeed believe the recursive reminders could be the mechanism underlying the effects outlined in Osgood’s proposal. We have modified the introduction in the following way (p. 5):

“Note that if increasing relatedness among word pairs along one or more dimensions increased RI, it would run contrary to Osgood’s predictions. Conversely, if increasing relatedness increased RF, it would support his predictions. Such results would also support recursive reminder theory (27), which we believe offers a mechanistic explanation of Osgood’s proposed surface because it predicts that retroactive benefits increase as reminders become more likely (such as with greater semantic relatedness). A further prediction of this theory is that relatedness would promote interdependence between associated memory traces.”

Moreover, both the recursive reminder hypothesis and Osgood's predictions deal with semantic relatedness. It is thus surprising that at the end of the Introduction, they were to examine several effects, such as the broad and narrow semantic association, the delay effect, the testing effect, and the effect of semantic relatedness on memory interdependency. Would the inclusion of these conditions help to falsify the hypothesis or predictions? The authors examined the semantic relatedness in different conditions separately but did not directly compare these conditions. They argued that this was because they were from different experiments. I am not sure if this argument is valid.

It is correct to note that manipulating the other parameters (delays, restudy vs. test, semantic associations) was not done to directly test the recursive reminders hypothesis. Rather, it was important for characterizing retroactive effects in the context of other known memory findings. That is, we manipulated delay because of its known importance for determining RF/RI (e.g., Chan, 2009); we manipulated restudy vs testing because of its prominence as a memory effect and the interesting interactions that semantics play in [such as the role of semantic mediators (e.g., Carpenter & Yeung, 2017)]; and we manipulated semantic associations to show that our initial effects using the narrower stimulus set were not solely due to high semantic relatedness in general.

It seemed very plausible that manipulating these factors would affect the shape of the surface (and indeed they did!). However, we believe the reviewer is pointing out that we did not directly manipulate recursive reminders, such as by altering instructions to think back in certain conditions like in Jacoby et al. (2015), and we agree. We have clarified this limitation and characterized it as a future direction in the Discussion (p. 25):

“Another limitation to our interpreting these results within a recursive reminders framework is that we did not directly manipulate instructions [e.g., (27)] or collect in-the-moment responses regarding whether subjects engaged in recursive reminders [e.g., (81)]. Such manipulations constitute important future directions.”

Regarding the last point, we avoided directly comparing results across experiments because we did not conduct random assignments to the different experiments. For example, we collected all 200 subjects for the retrieval-to-criterion, 48-hr, broad semantic association experiment before moving to the retrieval-to-criterion, 48-hr, narrow semantic association experiment. As a result, although we used the same subject population, we wanted to take a conservative approach to the statistics, just as experimenters often do when they discuss Experiment 1, Experiment 2, etc. without directly comparing their results via inferential statistics. However, we do note important qualitative differences in the results from the various experiments throughout the paper. We have clarified this point where we mention avoiding direct comparisons across experiments in the Discussion (p. 21):

“Although we have largely avoided comparing 5-min and 48-hr delay results directly because they were part of different experiments (and therefore subjects were not randomly assigned to different delays), it appears that conditions in which we expect recursive reminders tend to slow the rate of forgetting (relative to the amount of forgetting in the control condition).”

The authors introduce the testing effect to explain the delay effect and the difference between restudy and testing conditions. The testing effect, in my view, is mainly an observation, not an interpretation. Many possible mechanisms have been proposed and they are still under debate.

We agree there are many possible mechanisms for the testing effect. One of the most general observations is that testing benefits long-term memory relative to restudying, so we do think it is a relevant effect to discuss. We have changed the text to reflect the reviewer’s point about it being more of an observation and to reflect more openness to the testing effect debate (p. 21):

“These results mirror other phenomena that differentially affect short-term and long-term memory performance, such as the benefits of testing (characterized as the testing effect) (72, 103).”

and

“This pattern suggests recursive reminders act as effective retrieval processes, similar to findings on the testing effect (20). “

That said, I should note that this study examined a broad range of essential issues in RF and RI effects, and it is thus a challenge to come up with a simple theoretic framework to cover all of them. Meanwhile, I still think this study has provided critical empirical data for understanding the RF/RI effect. It is also laudable that they make the data open so that other researchers interested in this question could further examine it.

We thank the reviewer for these acknowledgements and for their excellent questions and critiques. Their efforts have substantially improved the manuscript.